# Temporal control of gene expression by the pioneer factor Zelda through transient interactions in hubs

Jeremy Dufourt [1], Antonio Trullo[1], Jennifer Hunter[1], Carola Fernandez[1], Jorge Lazaro[1], Matthieu Dejean[1], Lucas Morales[1], Saida Nait-Amer[1], Katharine N. Schulz [2], Melissa M. Harrison[2], Cyril Favard [3], Ovidiu Radulescu [4] & Mounia Lagha[1]

Pioneer transcription factors can engage nucleosomal DNA, which leads to local chromatin remodeling and to the establishment of transcriptional competence. However, the impact of enhancer priming by pioneer factors on the temporal control of gene expression and on mitotic memory remains unclear. Here we employ quantitative live imaging methods and mathematical modeling to test the effect of the pioneer factor Zelda on transcriptional dynamics and memory in *Drosophila* embryos. We demonstrate that increasing the number of Zelda binding sites accelerates the kinetics of nuclei transcriptional activation regardless of their transcriptional past. Despite its known pioneering activities, we show that Zelda does not remain detectably associated with mitotic chromosomes and is neither necessary nor sufficient to foster memory. We further reveal that Zelda forms sub-nuclear dynamic hubs where Zelda binding events are transient. We propose that Zelda facilitates transcriptional activation by accumulating in microenvironments where it could accelerate the duration of multiple pre-initiation steps.

[1] Institut de Génétique Moléculaire de Montpellier, University of Montpellier, CNRS-UMR 5535, 1919 Route de Mende, Montpellier 34293 cedex 5, France. [2] Department of Biomolecular Chemistry, University of Wisconsin School of Medicine and Public Health Madison, 6204B Biochemical Sciences Building 440 Henry Mall, Madison 53706 WI, USA. [3] Institut de Recherche en Infectiologie de Montpellier, CNRS, University of Montpellier UMR 9004, 1919 route de Mende, Montpellier 34293 cedex 5, France. [4] DIMNP, UMR CNRS 5235, University of Montpellier, Place E. Bataillon – Bât. 24 cc 107, Montpellier 34095 cedex 5, France. These authors contributed equally: Jeremy Dufourt, Antonio Trullo. Correspondence and requests for materials should be addressed to M.L. (email: mounia.lagha@igmm.cnrs.fr)

During the first stages of metazoan embryogenesis, the zygotic genome is largely quiescent, and development relies on maternally deposited mRNAs and proteins. Transcriptional activation of the zygotic genome occurs only hours after fertilization and requires specific transcription factors. In *Drosophila*, the transcriptional activator Zelda (Zld) plays an essential role in zygotic genome activation[1].

Zelda exhibits several pioneer factor properties such as (i) priming *cis*-regulatory elements prior to gene activation, which establishes competence for the adoption of cell fates, and (ii) opening chromatin, which facilitates subsequent binding by classical transcription factors. Indeed, hours before their activation, Zelda binds to thousands of genomic regions corresponding to enhancers of early-activated genes[2,3]. Zelda binding is associated with modifications of the associated chromatin landscape and overall chromatin accessibility[2–5].

Moreover, this ubiquitously distributed activator is able to potentiate the action of spatially restricted morphogens (*e.g* Bicoid and Dorsal)[6,7]. Consequently, target gene response is strengthened by Zelda binding, both spatially and temporally, for developmental enhancers[8] as well as for synthetic enhancers where input parameters are tightly controlled[9].

In addition to these properties, classical pioneer factors bind nucleosomes and are typically retained on the chromosomes during mitosis[10,11]. Mitotic retention of these transcription factors places them as ideal candidates for the transmission of chromatin states during cellular divisions, through a mitotic bookmarking mechanism.

However, the putative function of Zelda in mitotic memory has thus far not been examined. Moreover, all studies concerning Zelda's role as a quantitative timer have been performed on fixed embryos with limited temporal resolution. Thus, the effects of Zelda on the temporal dynamics of transcriptional activation and on transmission of active states through mitosis remain to be addressed.

Here we employ quantitative live imaging methods to directly measure the impact of Zelda on transcriptional dynamics in vivo. We show that increasing the number of Zelda-binding sites at an enhancer fosters temporal coordination in de novo gene activation (synchrony), regardless of past transcriptional states. By monitoring transcription in Zelda-maternally depleted embryos, we reveal that Zelda is dispensable for mitotic memory. On the contrary, we found that Zelda allows the temporal disadvantage of arising from a transcriptionally inactive mother to be bypassed. Using a mathematical modeling framework, we propose that mitotic memory requires long-lasting transitions, which are accelerated by Zelda, thus overriding mitotic memory of silent states. By analyzing Zelda protein kinetics during many cell cycles, here we report that while it is not retained on chromosomes during mitosis, this transcription factor exhibits a highly dynamic behavior. We observe that intra-nuclear distribution of Zelda is not homogeneous, and that it accumulates in local hubs, where its residence time is surprisingly transient (in the second range). Taken together, our results provide insights into Zelda-mediated temporal control of transcription during zygotic genome activation in vivo.

## Results and Discussion

**Zelda fosters temporal transcriptional coordination.** In the blastoderm embryo, two enhancers control *snail* (*sna*) gene expression, a proximal (primary) and a distal (shadow enhancer)[12,13]. Both enhancers are bound by Zelda (Zld) at early nuclear cycles (nc)[2,3]. When driven by its native enhancers, endogenous expression of *snail* is extremely rapid, precluding analysis of the impact of Zelda on fine-tuning the timing of its activation. We

therefore used a previously described truncated version of the *sna* shadow enhancer (*snaE*) that leads to a stochastic activation, compatible with the tracking of transcriptional memory across mitosis[14].

To follow transcriptional dynamics in a quantitative manner, we created a series of *enhancer* < *promoter* < *MS2-yellow* transgenes, with a unique minimal promoter (*sna* promoter) and an *MS2-yellow* reporter (Fig. 1a). Upon transcription, MS2 stem-loops are rapidly bound by the maternally provided MCP-GFP, which allows tracking of transcriptional activation in living embryos[15]. To decipher the role of Zelda in *cis*, a variable number of canonical Zld-binding sites (CAGGTAG)[16] were added to the *snaE* in three locations, either close to the promoter (Zld 3'), in the middle (Zld mid) or at a distance (Zld 5') (Fig. 1a).

Using fluorescent in situ hybridization, we confirmed that the pattern of expression of *snaE-extraZld* transgenes was dictated by the regulatory logic of the *sna* enhancer, and expression was indeed evident in the presumptive mesoderm, similarly to the endogenous *sna* pattern (Fig. 1b–g). To precisely measure temporal dynamics, we took advantage of our MCP/ MS2 system to track transcriptional activation in living embryos.

First, we implemented an automatic image analysis pipeline that segments nuclei in nc14 (Fig. 1h, i) and tracks the time of transcriptional activation (Fig. 1j, k). In the early fly embryo, a subset of transcriptional activators, including Dorsal, control gene expression in a graded manner[17]. It was therefore important to quantify temporal dynamics of gene activation in a spatially controlled domain. For this purpose, we used the ventral furrow (site of invagination during gastrulation process) as a landmark to define dorso-ventral coordinates (Fig. 1l, m and Supplementary Fig. 1a). Unless otherwise indicated, we studied temporal dynamics of gene activation in a region of 50μm centered around the ventral furrow to ensure non-limiting levels of the *sna* activators Dorsal (Supplementary Fig. 1b) and Twist[18].

We initially examined the impact of Zelda on the timing of gene activation. As identified with static approaches[6,7,16], our dynamic data showed that increasing the number of Zld-binding sites resulted in precocious transcriptional activation (Fig. 1n–q Supplementary Fig. 1c, d and Supplementary Movie 1-4). Zelda-binding impacted not only the onset of gene activation but also the temporal coordination among a spatially defined domain (*i.e* the presumptive mesoderm, 50μm around the ventral furrow), referred to as synchrony[19] (Fig. 1r and Supplementary Fig. 1d). For example, at precisely 10 min into interphase 14, expression of the *snaE* transgene was not coordinated (≈16% of active nuclei), whereas the number of active nuclei at this stage increased upon the addition of extra Zld-binding sites (Fig. 1n–q). The precise kinetics of gene activation, *i.e* synchrony curves, was quantified as a percentage of active nuclei within a defined spatial domain for each transgene during the first 30 min of nc14. SnaE generated a slow activation kinetic, where half of the domain is activated at t50≈18 min (Fig. 1r). Adding only a single CAGGTAG sequence, 5' to the enhancer accelerated this kinetic (t50≈13 min). With two extra Zld-binding sites synchrony is further increased (t50≈9.5 min). The activation kinetics of two extra canonical Zld sites mimicked the dynamics generated by the intact long shadow enhancer (t50≈9.5 min) (Fig. 1r). The most dramatic synchrony shift was caused by the addition of a single CAGGTAG sequence close to the promoter ( + 1Zld 3') (t50≈6.5 min) (Supplementary Fig. 1d). This behavior was not changed when two additional Zelda sites were added, suggesting that promoter proximal Zelda binding causes a strong effect, that cannot be further enhanced by the addition of more distant sites (Fig. 1r and Supplementary Fig. 1d). By moving the CAGGTAG site away from the promoter, to the middle of the enhancer, synchrony was decreased (t50≈7.5 min) (Supplementary Fig. 1d)

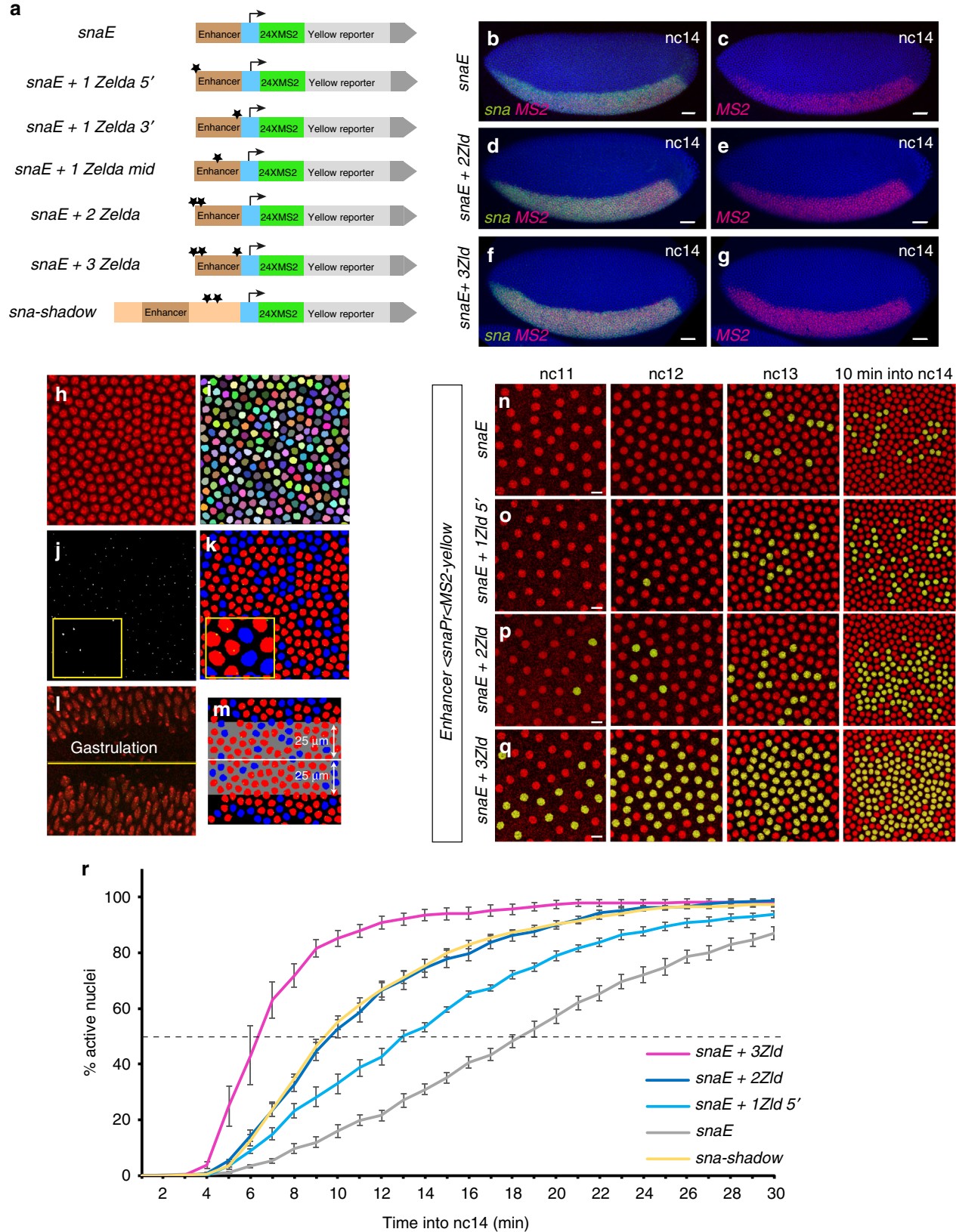

**Fig. 1** Zelda-binding sites provide quantitative temporal control of gene expression. **a** Schematic representation of the truncated version of the *sna* shadow enhancer (*snaE*) and the variants carrying different numbers and locations of extra Zelda (Zld)-binding sites (black stars represent canonical Zld-binding sites: CAGGTAG). Each enhancer controls a common minimal promoter (blue box, *sna* promoter) that transcribes a *yellow* gene with 24 *MS2* repeats located at its 5' end. **b–g** Transgenic embryos stained with a *MS2* probe (red, panels **b–g**) and with a *snail* probe (green, **b**, **d**, **f**). Dorsal views at nc14, anterior to the left, scale bars represent 30μm. **h–m** Workflow for automatic tracking of single nucleus transcriptional activity. **h**, **i** Snapshot of a maximum intensity projected Z-stack from live imaging movie representing nuclei labeled with *His2Av-mRFP* transgene (**h**) and associated automatic segmentation, where each tracked nucleus is given a random color (**i**). **j** Maximum intensity projected Z-stack image extracted from a typical movie showing the transcriptional foci (white dot). **k** False-colored image, representing the segmented transcriptionally inactive nuclei (blue), active nuclei (red) and associated transcriptional foci (yellow dot). **l** Maximum intensity projected Z-stack image representing a typical gastrulation, used as a landmark to determine dorso-ventral (D/V) coordinates (yellow line). **m** Representative image exhibiting the spatial domain (grey, here 25μm surrounding the ventral furrow) defined by precise D/V coordinates. **n–q** Live imaging of transcriptional activity driven by the *snaE* upon increasing numbers of Zld-binding sites. Nuclei that displayed transcriptional activity were false-colored in yellow. Snapshots from maximum intensity projected Z-stack of live imaging movies are shown from nc11 to nc14. Scale bars represent 10μm. **r** Temporal coordination profiles during nc14. Synchrony curves were obtained by quantifying transcriptional activation during nc14 in a region of 50μm around the ventral furrow: *snaE* (6 movies, $n = 970$ nuclei) (grey curve), *snaE + 1Zld5'* (4 movies, $n = 700$ nuclei) (light blue curve), *snaE + 2Zld* (5 movies, $n = 723$ nuclei) (dark blue curve), *snaE + 3Zld* (4 movies, $n = 824$ nuclei (purple curve)) and *snaE-shadow* (4 movies, $n = 612$) (yellow curve). Dashed line represents 50% of activation. The time origin is the end of mitosis (start of nc14). Error bars represent SEM

suggesting a stronger role for Zelda when positioned close to the promoter.

In this system, addition of a single CAGGTAG was sufficient to increase the activation kinetics. This is in sharp contrast to what has been reported for a synthetic enhancer, whereby activation only occurred when three or more Zelda-binding sites were added[9]. This difference is likely due to the ability of Zelda to function cooperatively with other activators of *sna*, which are not present in the synthetic enhancer previously examined[9].

While our initial analysis focused on the region surrounding the ventral furrow, live imaging data also provided insights into the temporal activation in the entirety of the *sna* pattern. By temporally tracking activation along the dorso-ventral axis, we could observe the consequences of the Dorsal nuclear spatial gradient[20] (Supplementary Fig. 1e). Indeed, the first activated nuclei were located where activators are at their peaks levels, whilst the late activated nuclei were closer to the mesodermal border, as previously suggested by analysis of fixed embryos and modeling[7,21]. By comparing activation timing between 0-50μm from the ventral furrow, we show that adding extra Zld-binding sites led to more rapid activation across the entire spatial domain (Supplementary Fig. 1e). These temporal data are consistent with the idea that Zelda potentiates the binding of spatially restricted morphogens like Bicoid or Dorsal[6–8,22] and thus support the previously described pioneering activities of Zelda[5].

**Zelda bypasses transcriptional memory**. Our ability to track transcriptional activation in live embryos provided the opportunity to determine activity through multiple nuclear cycles. Using the sensitized *snaE* transgene, we recently documented the existence of a transcriptional mitotic memory, whereby the transcriptional status of mother nuclei at nc13 influences the timing of activation of the descendants in the following cycle[14]. To address the challenge of following rapid movements during mitosis, we developed a semi-automatic mitotic nuclei tracking software (Fig. 2a). By combining lineage information with transcriptional status, we quantified the timing of activation of hundreds of nuclei in nc14 in a spatially defined domain. This allowed us to distinguish those transcriptionally active nuclei that arose from an active mother nucleus (green curves, Fig. 2b–d and Supplementary Fig. 1f, g) from those coming from an inactive mother nucleus (red curves, Fig. 2b–d and Supplementary Fig. 1f, g). Differences between the kinetics of activation of these two populations provide an estimation of transcriptional mitotic memory (Fig. 2b).

Based on the ability of Zelda to define hundreds of cis-regulatory regions through the establishment or maintenance of accessible chromatin[4,5], we reasoned that Zelda could be in part responsible for the establishment of transcriptional mitotic memory during early development. Thus, we leveraged our live imaging system to directly test the role of additional Zld-binding sites on transcriptional activation through mitosis. Contrary to our expectation, namely that extra Zelda transgenes would exhibit enhanced memory; adding extra Zld-binding sites in *cis* augmented the speed of activation in both nuclei derived from active mothers and those from inactive mothers (Fig. 2c, d and Supplementary Fig. 1f, g). Increasing the number of Zld-binding sites reduced the differences in the timing of activation for descendants from active mothers and their neighboring nuclei, arising from inactive mothers. Our results suggest that extra Zld-binding sites accelerate transcriptional dynamics regardless of the past transcriptional status. Thus, Zelda seems capable of bypassing transcriptional memory by compensating for the negative effect of having an inactive mother nucleus.

**Zelda is dispensable for transcriptional mitotic memory**. We then sought to test whether Zelda was required for this memory. Activation kinetics for descendants from active and inactive nuclei were quantified after reducing maternal Zelda expression using RNAi. Validating this approach, germline expression of *zld*-RNAi[7,23] reduced the levels of *zld* mRNA to 2–4% of that found in control *white*-RNAi embryos (Supplementary Fig. 3a).

Reduction of maternal Zelda decreased the synchrony of activation of our transgenes with additional Zld-binding sites (Fig. 3a–j, Supplementary Fig. 3b–d and Supplementary Movie 5, 6). In *zld*-RNAi embryos, temporal coordination was reduced, and all transgenes showed a t50≈16–19 min (Fig. 3i, j, Supplementary Fig. 3b-d and Supplementary Movie 5, 6). This confirms that the previously identified acceleration of transcriptional activation upon addition of extra Zld-binding sites is due to Zelda activity on these enhancers. Further confirming the role of maternally contributed Zelda in potentiating Dorsal-dependent gene expression, transcriptional activation was restricted to a reduced region of about ≈25 μm around the pseudo-furrow in Zelda maternally depleted embryos (Supplementary Movie 6). Together, these data demonstrate that expression of our *snaE* transgenes reflect the general properties of Zld-mediated gene expression.

We then quantified the level of memory in the *zld*-RNAi embryos with three *snaE < MS2* transgenes containing varying number of Zld-binding sites. In spite of the maternal reduction of

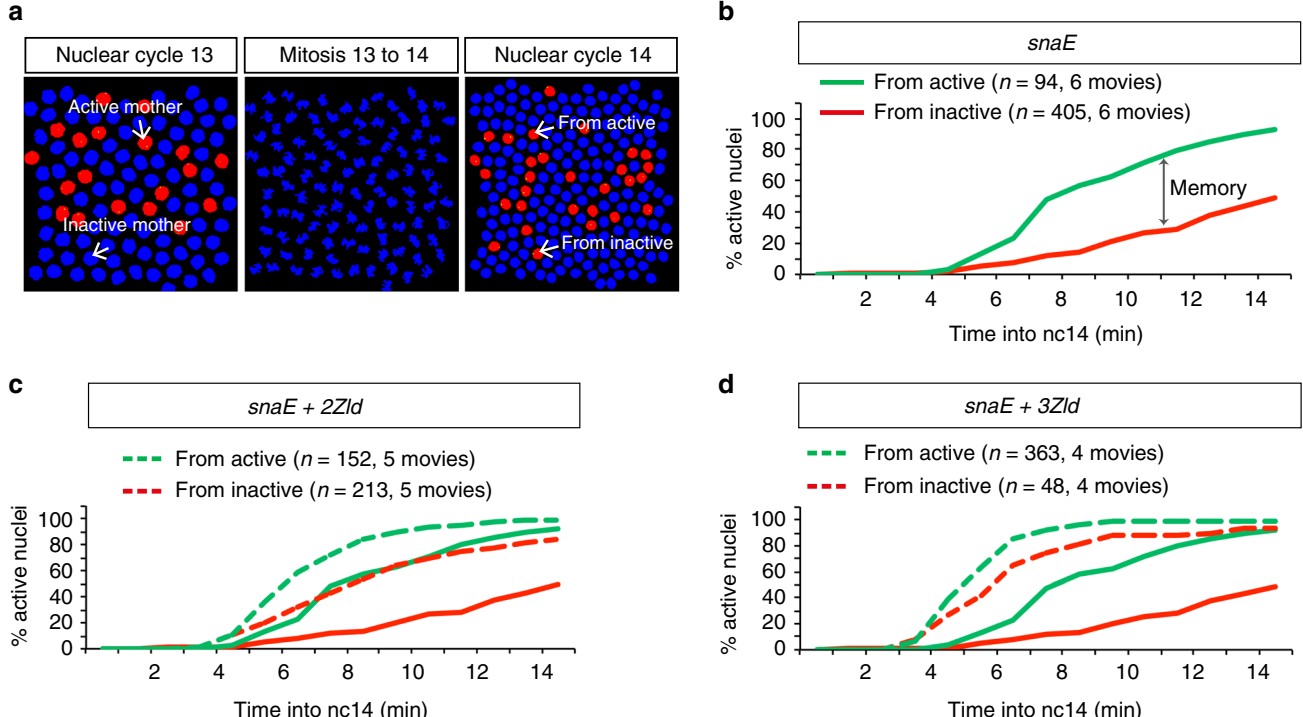

**Fig. 2** Zelda accelerates gene activation regardless of transcriptional history **a** Representative images from the Graphical User Interface implemented for automatic tracking of nuclei across multiple divisions. Chromosomal splitting at the end of anaphase is automatically detected for each nucleus. **b–d** Quantification of transcriptional memory in various *snaE < MS2* embryos. At each time point in nc14, the cumulative number of activated nuclei, expressed as a percentage of the total number of nuclei, is plotted after segregation into two populations: descendants of active mothers (green) and descendants of inactive mothers (red). The timing of activation of each nucleus has been normalized to its specific timing of mitosis. Temporal dynamics from *snaE* are depicted using solid curves, while those from transgenes with extra Zld-binding sites are represented as dashed curves (**c**, **d**)

Zelda, a strong memory bias was still observed (Fig. 3k, l and Supplementary Fig. 3e). We conclude that although possessing several key properties of a pioneer factor, Zelda is not necessary to elicit heritability of active transcriptional states through mitosis.

Altogether our results show that Zelda accelerates transcription and fosters synchrony, despite playing no role in transcriptional memory maintenance.

To further examine whether Zelda binding to an enhancer was sufficient to trigger a memory bias, temporal activation from a second regulatory element (*DocE*) was examined (Supplementary Fig. 2). This region is located in a gene desert within the *Doc1/2* locus and is characterized by high Zelda occupancy in early embryos (Supplementary Fig. 2a, b)[3]. We created an *MS2* reporter transgene with this putative enhancer and found that it drives expression in the dorsal ectoderm in a pattern overlapping endogenous *Doc1* mRNA expression (Supplementary Fig. 2c, d). Upon *zld*-RNAi, expression from this enhancer was strongly reduced, confirming a role for Zelda in driving expression from the *DocE* element (Supplementary Movie 7 and 8).

During nc13, our *DocE* transgene was stochastically activated in three distinct dorsal domains, in which we could track transcriptional mitotic memory (Supplementary Fig. 2c). None of the three domains (anterior, central, posterior) revealed a bias in the timing of activation in nc14 for descendants of active mothers compared to those arising from inactive mothers (Supplementary Fig. 2e–h). We therefore conclude that this *DocE* element does not trigger memory, despite being bound by Zelda. Thus, in a model system that shows stochastic activation and that uses Zelda, we did not observe any mitotic memory, further confirming the lack of involvement of Zelda in memory.

Moreover these results show that experiencing transcription at a given cycle does not necessarily lead to a rapid *post*-mitotic reactivation in the following cycle.

**Modeling the impact of Zelda on synchrony and memory**. To gain insights into the role of Zelda on transcriptional dynamics, we developed a simple mathematical framework that describes the waiting times prior to the first detected transcriptional initiation event in nc 14, as a sequence of discrete transitions (see Methods and Supplementary Methods). This model considers that the activation of transcription follows a stepwise progression through a series of nonproductive OFF events that precede activation (ON event) (Fig. 4a). Given the various well characterized steps required prior to productive transcriptional elongation (e.g. promoter opening, transcription factor binding, pre-initiation complex recruitment...), it is reasonable to consider a series of transitions that a promoter must travel through prior to activation, with an allocated duration[11,24].

In this model, the average number of rate-limiting transitions is provided by the parameter '*a*' while the duration of the transitions by parameter '*b*' (expressed in seconds and considered the same for all transitions, a more complex model with heterogeneous '*b*' is discussed in the Supplementary Methods) (Fig. 4a). In the model, the distribution of waiting times prior to activation for a given gene in a set of nuclei can be described as a mixture of gamma distributions. Gamma distributions are frequently used in statistics for modeling waiting times. These distributions depend on two parameters, the shape parameter '*a*' and the scale parameter '*b*'. When, like in our model, the waiting time is the sum of a number of independent, exponentially

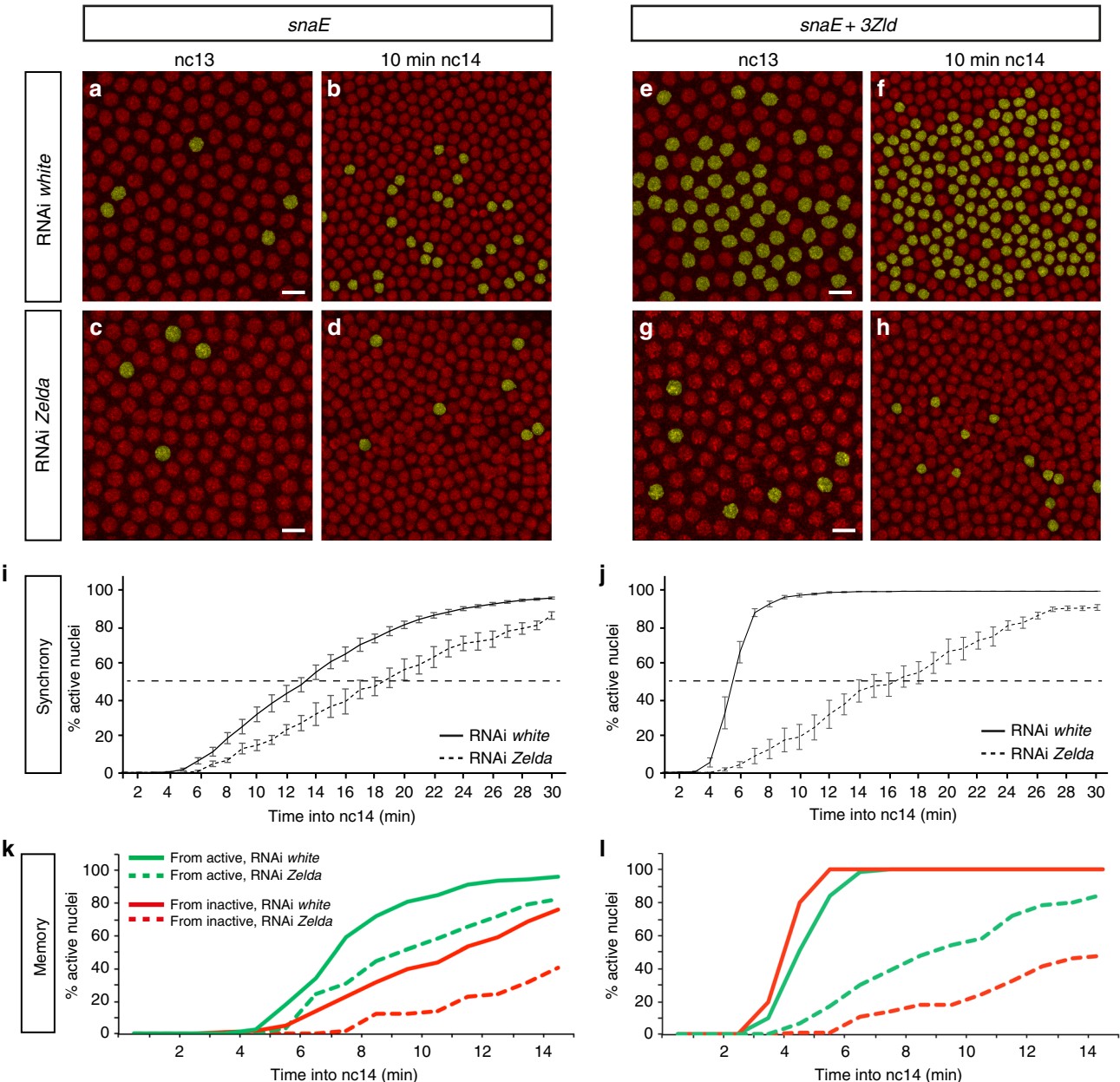

**Fig. 3** Transcriptional memory persists in the absence of Zelda. **a–h** Snapshots of maximum intensity projected Z-stack of *Drosophila* embryo movies expressing *snaE* (**a–d**) or *snaE + 3Zld* (**e–h**) at nc13 (**a, c, e, g**) and nc14 (**b, d, f, h**), in a *white*-RNAi (**a, b, e, f**) or in a *zld*-RNAi genetic background (**c, d, g, h**). Nuclei exhibiting a spot of active transcription are false-colored in yellow. **i, j** Temporal coordination of all activated nuclei in nc14 for *snaE* (**i**) and *snaE + 3Zld* transgene (**j**), in a *white*-RNAi (solid curves) and in a *zld*-RNAi genetic background (dashed curves). Statistics: *snaE white*-RNAi: 9 movies, n = 972; *snaE zld*-RNAi: 4 movies n = 156; *snaE + 3Zld white*-RNAi: 6 movies n = 541; *snaE + 3Zld zld*-RNAi: 5 movies n = 160. Error bars represent SEM. **k, l** Kinetics of activation during the first 15 min of nc14 driven by the *snaE* transgene or the *snaE + 3Zld* transgene in *white*-RNAi (solid curves) or in *zld*-RNAi embryos (dashed curves). The kinetics of nuclei coming from transcriptionally active mothers in nc13 (green curves) is compared to those arising from inactive mothers (red curves). Statistics: *snaE white*-RNAi: 6 movies, n = 302; *snaE zld*-RNAi: 4 movies, n = 86; *snaE + 3Zld white*-RNAi: 6 movies, n = 273; *snaE + 3Zld zld*-RNAi: 5 movies n = 111. In Fig. 3, only nuclei located within a 25 μm rectangle centered around the ventral furrow/pseudo-furrow for *zld*-RNAi are considered. Scale bars represent 10 μm

distributed steps of equal mean duration, 'a' is the number of transitions (steps) while 'b' is the mean duration. Thus, 'a' = 1 corresponds to the exponential distribution. A mixture of gamma distributions covers the case when the number of transitions (parameter 'a') is random. The cumulative distribution function for the mixture is formulated in Fig. 4a and in the Supplementary Methods. Using the model, the average number of steps (the mean parameter 'a' in the mixture) can be computed from the

first two moments of the distribution of waiting times (Supplementary Equation 3 Supplementary Methods). This rough but direct estimate shows that 'a' is less than three for all genotypes. Thus, we have restricted our model to three limiting transitions and consequently to three nonproductive states (OFF1, OFF2 and OFF3) (Fig. 4a).

Taking advantage of the significant number of nuclei tracked in this study, we could fit this model to our data and estimate more

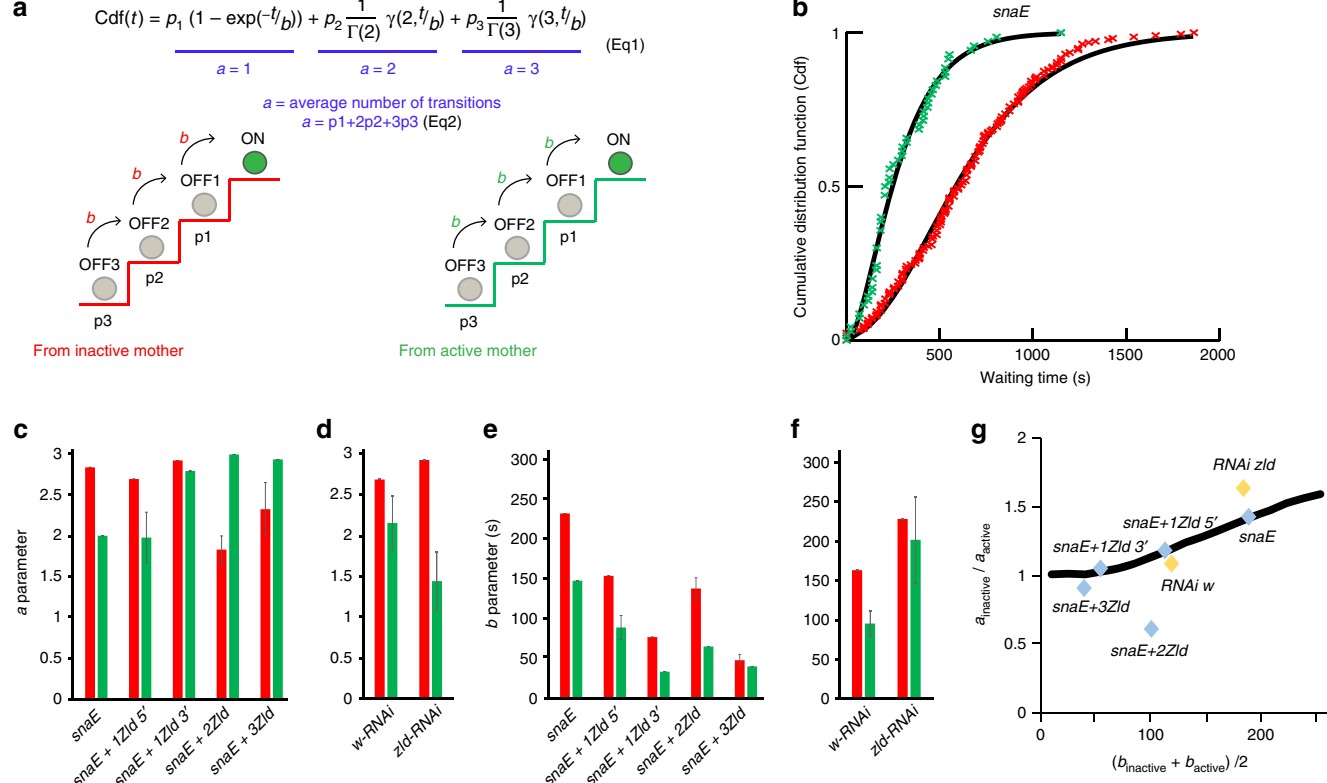

**Fig. 4** Mathematical modeling of transcriptional memory. **a** Schematic representation of the discrete states (OFF1, OFF2 and OFF3) required to reach transcription (ON state), when coming from an inactive (red stairs) or an active (green stairs) mother nucleus. After mitosis, the probability for a nucleus to initiate the transcriptional activation process from a given state is provided by the parameter $p$, ($p_1 = $p(OFF1), $p_2 = $p(OFF2) and $p_3 = $p(OFF3)). The distribution of the random time to transcription ($t$) is modeled as a mixture of gamma distributions, according to the displayed Equation 1, where Cdf is the cumulative distribution function, $\Gamma$ and $\gamma$ are the complete and incomplete gamma functions, respectively. The average number of transitions '$a$' is provided by the sum of weighted probabilities (Equation 2). The time of each jump from one state to another is provided by the parameter '$b$'. Parameters of this model are predicted separately for nuclei coming from active and inactive mothers. The parameter fitting is based on Cdf of the random time to transcription in nc14 ($t$) (see also Supplementary Equation 1 Supplementary Methods). **b** Cdf of the random time to transcription in nc14. The origin of time is the end of mitosis nc13 to nc14, (determined by our automatic software, see "Methods" section) added to the time required to detect the first activation (proper to each genotype). Data (cross) are compared to predictions (solid curves) for nuclei, resulting from active (green) and inactive (red) mothers. **c**, **d** Mean '$a$' parameter; error intervals correspond to the variation among optimal and best suboptimal fits (see Supplementary Methods and Supplementary Data 1). **e**, **f** Mean '$b$' parameter; error intervals correspond to variation among optimal and best suboptimal fits (see Supplementary Methods and Supplementary Data 1). **g** The ratio of parameter '$a$' in subpopulations from inactive and active mothers is positively correlated (Spearman correlation coefficient = 0.82 $p = 0.034$) to the average '$b$' parameter. Predictions of the mathematical model are represented as a black line

accurately the parameters '$a$' and '$b$', from populations of nuclei (descendants of active and descendants of inactive nuclei) (Supplementary Data 1).

The distribution of waiting times for descendants of active nuclei and inactive nuclei were clearly distinct, indicating distinguishable scale and average shape parameters for these two populations (Fig. 4b). For *snaE*, the temporal behavior of descendants of active and inactive mothers differed in both parameters '$a$' and '$b$', with both being lower for descendants of active nuclei (Fig. 4c, e). Thus, our model suggests that descendants of active nuclei have fewer and shorter transitions prior to activation than descendants of inactive nuclei.

For transgenes with additional Zld-binding sites, the predictions for '$a$' are not obviously affected (Fig. 4c). This suggests that Zelda binding does not dramatically alter the number of transitions required for activation. By contrast, the estimates for '$b$' are systematically lower for descendants of active mothers and inactive mothers upon the addition of Zld-binding sites (Fig. 4e).

Thus, while the accelerated transcriptional activation observed with increased numbers of Zld-binding sites could be due to

either a decrease in the duration of the transitions '$b$' or to the number of transitions '$a$', our modeling suggests that Zelda accelerates transcriptional activation primarily by speeding the lifetime of the transitions. Consistent with these findings, upon maternal reduction of Zelda, the '$b$' parameter is augmented, when compared to estimates in *white RNAi* controls (Fig. 4f).

In contrast to nuclei expressing the *snaE* transgene, the relation between estimates of '$a$' for descendants of active mothers and those of inactive mothers is more complex for extra-Zld transgenes (Fig. 4c, d). In order to gain some insights, we have used numerical simulations of an extended version of our model that includes modified transition dynamics during mitosis (Supplementary Fig. 4). In this version, we consider that at the beginning of mitosis, the states of active and inactive mother nuclei are OFF1 and OFF3, respectively. During mitosis, nuclei can undergo reversible (upward and backward) transitions. After mitosis, the resulting daughter nuclei follow the irreversible transition scheme represented in Fig. 4a. The simulations predict that the bias in '$a$' values (difference in the number of steps to reach active state, evaluated by $a_{inactive}/ a_{active}$ and referred to as

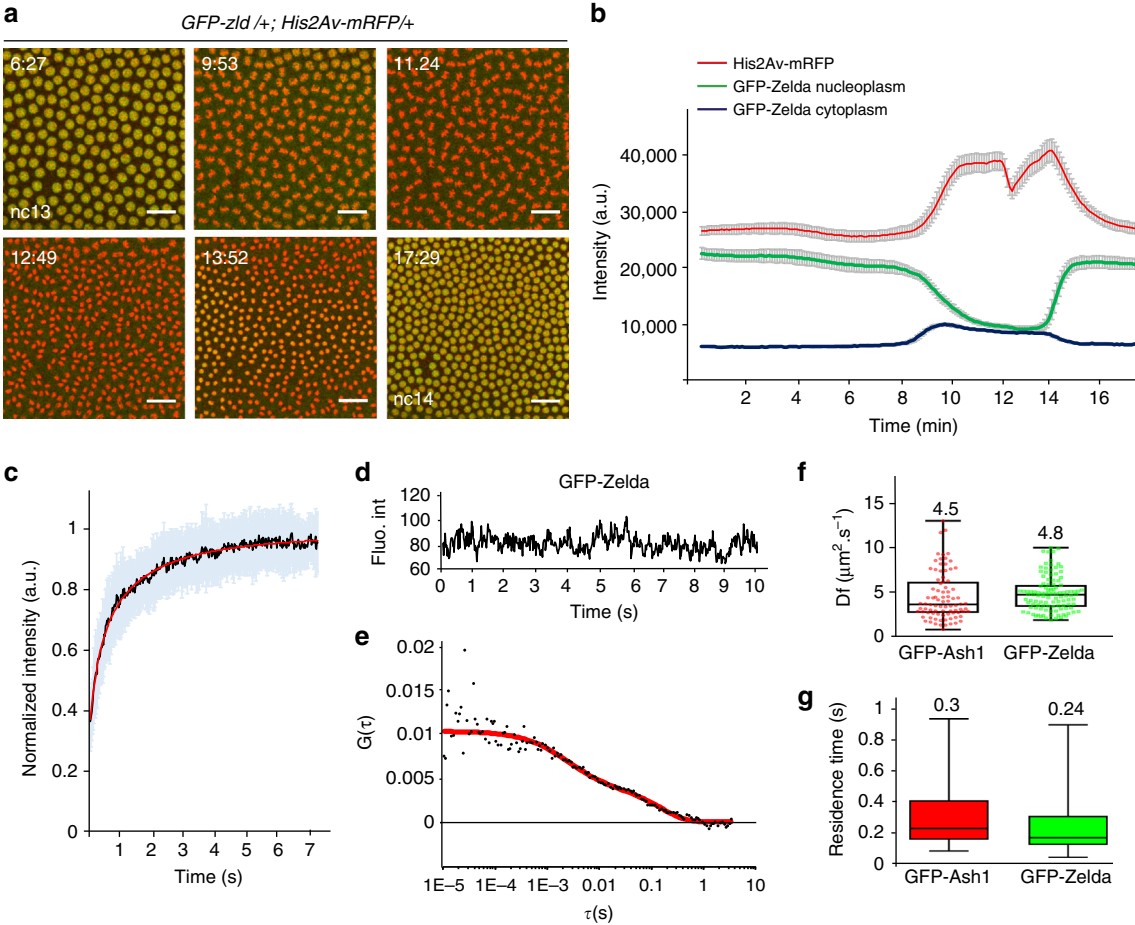

**Fig. 5** Zelda global kinetic properties. **a** Living *GFP-zld/+ ;His2Av-mRFP/+* embryo imaged by confocal microscopy from interphase of nc13 to early interphase of nc14. Successive representative maximum intensity projected Z-stack images are shown at the indicated timings (in minutes) (see also Supplementary Movie 9). Scale bars represent 20 μm. **b** Average intensity profiles for histones (red), nucleoplasmic Zelda (green) and cytoplasmic Zelda (blue) measured from a nc13 embryo transitioning into nc14. An automatic tracking of fluorescence from a minimum of 114 nuclei and 10 cytoplasmic ROI generated these profiles. Error bars represent SD. **c** FRAP mean curve (black) and the mean of all the fits (red curve) using reaction-diffusion models determined at the bleached spot for 25 nuclei from nc14 developing *GFP-zld* embryos. Error bars represent SD from different nuclei (light blue bars). **d** Example of a time trace obtained by FCS from a *GFP-zld* nc14 embryo showing no bleaching. **e** Example of autocorrelation function (black dots) related to **d** (red curve represents fitting using a reaction-diffusion model). **f**–**g** Kinetic parameters for Zelda and Ash1 extracted after fitting FCS data with a reaction-diffusion model. **f** Box plot representing estimated diffusion coefficients ($D_f$). Numbers above each plot represent the mean. Centered lines represent the median and whiskers represent min and max. **g** Box plot representing estimated residence times ($1/k_{off}$). Numbers above each plot represent the mean. Centered lines represent the median and whiskers represent min and max

memory bias) is correlated to the '*b*' values (Supplementary Methods). Remarkably, we could obtain this correlation with our temporal data (Fig. 4g). Hence, enhancers with large '*b*' values (having slow transitions) tend to have a strong $a_{inactive}/a_{active}$ ratio, whereas enhancers with small '*b*' values have a vanishing memory bias (Fig. 4g and Supplementary Methods). If we interpret transcriptional memory as the heritable reduction in the number of transitions required to reach the active state, this result suggests that stronger memory occurs when transitions are slower.

In summary, using a modeling approach, we converted our measurements of the timing of transcriptional activation in hundreds of single nuclei into discrete metastable states preceding gene activation. Although purely phenomenological, this model suggests that transcriptional memory is supported by a sequence of relatively long-lived metastable states preceding activation, which could be maintained by mitotic bookmarking. Zelda potentiates transcription by accelerating these states and is thus incompatible with mitotic memory, consistent with our observations in Zelda depleted embryos.

**Zelda dynamic behavior**. Based on the characteristics of pioneer factors, we had expected a role for Zelda in retaining transcriptional memory through mitosis. However, our genetic data and modeling indicate that Zelda was not the basis of memory. To better understand this result, we examined Zelda dynamics in living embryos particularly during mitosis (Fig. 5a, b and Supplementary Movie 9). We took advantage of an endogenously tagged fluorescent version of Zelda[25]. The *GFP-zld* flies are homozygous viable, thereby showing that the GFP-tagged version of Zelda retains wild-type physiological properties.

GFP-Zld localization with chromatin was examined through the mitotic cycle (Fig. 5, Supplementary Fig. 5 and Supplementary Movie 9) and demonstrated that Zelda is not detectably retained on the chromosomes during mitosis (Fig. 5a, b and Supplementary Movie 9), similar to what was shown in fixed embryos[26]. This is in comparison to the transcription factor GAF (GAGA Associated Factor) (Supplementary Fig. 5a). While most GFP-Zld molecules are localized in nuclei in interphase, they were redistributed to the cytoplasm during mitosis (Fig. 5b). Our live imaging approach revealed a highly dynamic behavior of Zelda,

whereby Zelda comes back to the nucleus rapidly at the end of mitosis prior to complete chromosome decompaction (Fig. 5a, b, Supplementary Fig. 5b and Supplementary Movie 9). When compared to Pol II-Ser5P by immunostaining on embryos exhibiting a mitotic wave, GFP-Zld signal filled the nuclei prior to polymerase (Supplementary Fig. 5b). By comparing Zelda cell cycle dynamics to Bicoid[27], we found that these two DNA binding proteins are similarly evicted during mitosis and are quickly re-localized to chromatin at the end of mitosis (Supplementary Fig. 5c and Supplementary Movie 10). Nonetheless, quantitative comparisons revealed that Zelda entry into the nucleus at the end of mitosis was faster than that of Bicoid (Supplementary Fig. 5d).

To better characterize Zelda kinetics, we performed intranuclear Fluorescence Recovery After Photobleaching (FRAP) experiments and determined that Zelda recovery was remarkably fast in interphase nuclei (Fig. 5c). Because FRAP is not well suited for fast moving proteins, we performed Fluorescence Correlation Spectroscopy (FCS) in living cycle14 embryos (Fig. 5d, e) and estimated the kinetic properties of Zelda (Fig. 5f, g). FCS experiments for Zelda were compared to Ash1, whose kinetics have been documented by both FRAP and FCS in early *Drosophila* embryos[28] and thus serves as a well-defined point of reference (Supplementary Fig. 5e–g). As expected for chromatin-binding proteins, auto-correlation curves for Zelda and Ash1 show more than one unique characteristic time (Fig. 5e and Supplementary Fig. 5f). To estimate chromatin-binding kinetics, FCS curves (86 curves for Ash1 and 109 curves for Zelda) were fitted with a reaction-diffusion model[29]. As previously revealed in Steffen et al.[28], we found that Ash1 is highly dynamic and estimate similar diffusion coefficients ($D_f \approx 4.5 \mu m^2 \cdot s^{-1}$ versus $D_{f(Steffen)} \approx 4.98\ \mu m^2 \cdot s^{-1}$) (Fig. 5f), thus validating our FCS experiments. Similarly to other transcription factors (e.g. Bicoid) we found that Zelda is a fast diffusing protein with apparent diffusion coefficient of $\approx 4.8\ \mu m^2 \cdot s^{-1}$ (Fig. 5f).

We also extracted the dissociation rate ($k_{off}$) (see Methods) and deduced the estimated residence time (RT) (RT = $1/k_{off}$) (Fig. 5g). Our FCS results reveal that Zelda binding to chromatin is highly transient, on the order of hundreds of milliseconds (RT = 0.24 s). However, we note that this FCS-estimated RT likely represents only a lower-bound limit for Zelda binding, whereas a FRAP approach allows for upper-bound estimations (see next paragraph).

The low binding and fast diffusion rates for Zelda were unexpected given the large impact of a single extra Zld-binding site on accelerating the kinetics of gene activation (Fig. 1). However, similar transient chromatin bindings have been reported recently for other key transcription factors[8,30]. These transient bindings events seem to be compensated for by increased local concentrations, in particular nuclear microenvironments, referred to as subnuclear hubs[8]. We thus explored Zelda spatial distribution within nuclei in living *Drosophila* embryos.

**Zelda accumulates into subnuclear hubs.** Live imaging with GFP-Zld, revealed a heterogeneous distribution of Zelda proteins in the nucleus (Fig. 6a–d and Supplementary Movie 11–13). Thus, similarly to Bicoid[8], multiple Zelda proteins seem to cluster in local dynamic hubs that exhibit a diversity of sizes and lifetimes (Fig. 6a–b and Supplementary Movie 11–13). For example, Zelda large hubs are more visible at early cycles and at the beginning of interphases (Fig. 6a and Supplementary Movie 11, 13). These local inhomogeneities are unlikely GFP aggregate as we could observe them with a two-colored Zelda tagging approach (Fig. 6c). Consistent with this, Zelda hubs have been observed very recently with other methods[31].

These findings are consistent with a "phase separation" model of transcriptional regulation[32], which could promote Zelda cooperativity with other major transcriptional regulators, like Bicoid[8], which would then foster rapid and coordinated transcriptional initiation.

To decipher the link between Zelda hubs and transcription, we examined the spatial localization of nascent transcription with respect to Zelda hub positions.

Immuno-FISH and live imaging experiments revealed that only a subset of our MS2 reporter nascent RNAs were located within a Zelda dense region (Fig. 6d, Supplementary Fig. 6a, and Supplementary Movie 13). Furthermore, only limited co-localization was observed between dots of elongating Pol II (Ser2-P Pol II) and Zelda hubs (Supplementary Fig. 6b). Similar findings have been recently reported by following transcription from a *hunchback < MS2* transgene (*hb*)[31]. We therefore conclude that, at least with a synthetic transgene, long-lasting stable contacts between sites of transcription and Zelda dense regions are not detected. Given the important number of Zelda targets, rigorously connecting transcriptional activation to Zelda hubs would require a broader analysis with adapted imaging methods. However, these observations are reminiscent of a dynamic kissing model recently described between an active gene and mediator clusters in ES cells[33].

Having demonstrated that Zelda was not homogeneously distributed, we wanted to determine if the dynamic properties of Zelda within hubs differed from its global dynamics.

One possibility was that high local concentrations of Zelda would correspond to long-lasting chromatin-binding events that would result in increased residence time (long RT). Given the rapid movements of Zelda hubs (Supplementary Movie 11–14), capturing these long-lasting events by FCS was not feasible. Therefore, we performed FRAP experiments on long-lasting Zelda hubs (Fig. 6e and Supplementary Movie 14). We demonstrate that Zelda recovered as rapidly within long-lasting hubs as it did in bulk nuclei (Fig. 6f and Fig. 5c). By fitting our FRAP data with a diffusion-reaction model, we estimate a residence time on the order of seconds in both cases (hubs and global FRAP) (Fig. 6g). Fitting the FRAP experiments with alternative models, such as one binding reaction only (Supplementary Fig. 6c–e) or two distinct binding reactions (slow and fast) (Supplementary Fig. 6f–h) similarly supported an estimated residence time for Zelda on the order of seconds and that this is independent of its local concentration (Supplementary Fig. 6i). Of note, the two reactions model estimates two residence times, where the fastest estimate is similar to the FCS-extracted residence time (Fig. 5g and (Supplementary Fig. 6i)). Previous studies on Bicoid have also identified two populations: one with a short characteristic time (RT on the order hundreds of milliseconds), described as non-specific binding events and a population with longer characteristic times (RT on the order of seconds) corresponding to specific binding events[8,34].

From the combined data, we conclude that Zelda chromatin binding is transient, ranging from hundreds of milliseconds (likely unspecific binding) to seconds (likely specific binding) and this occurs regardless of its local concentration. How could Zelda exhibit a similarly transient chromatin binding at random positions and in regions where its concentration is high? An exciting possibility would be that dense regions could be enriched for Zelda target genes. Under this hypothesis, Zelda target genes could cluster together thereby increasing the probability to locally accumulate Zelda proteins. Alternatively, Zelda could spatially organize its targets in space through its pioneer factor properties and then maintain these clusters of targets possibly through its intrinsically disordered domains[35]. The latter scenario is

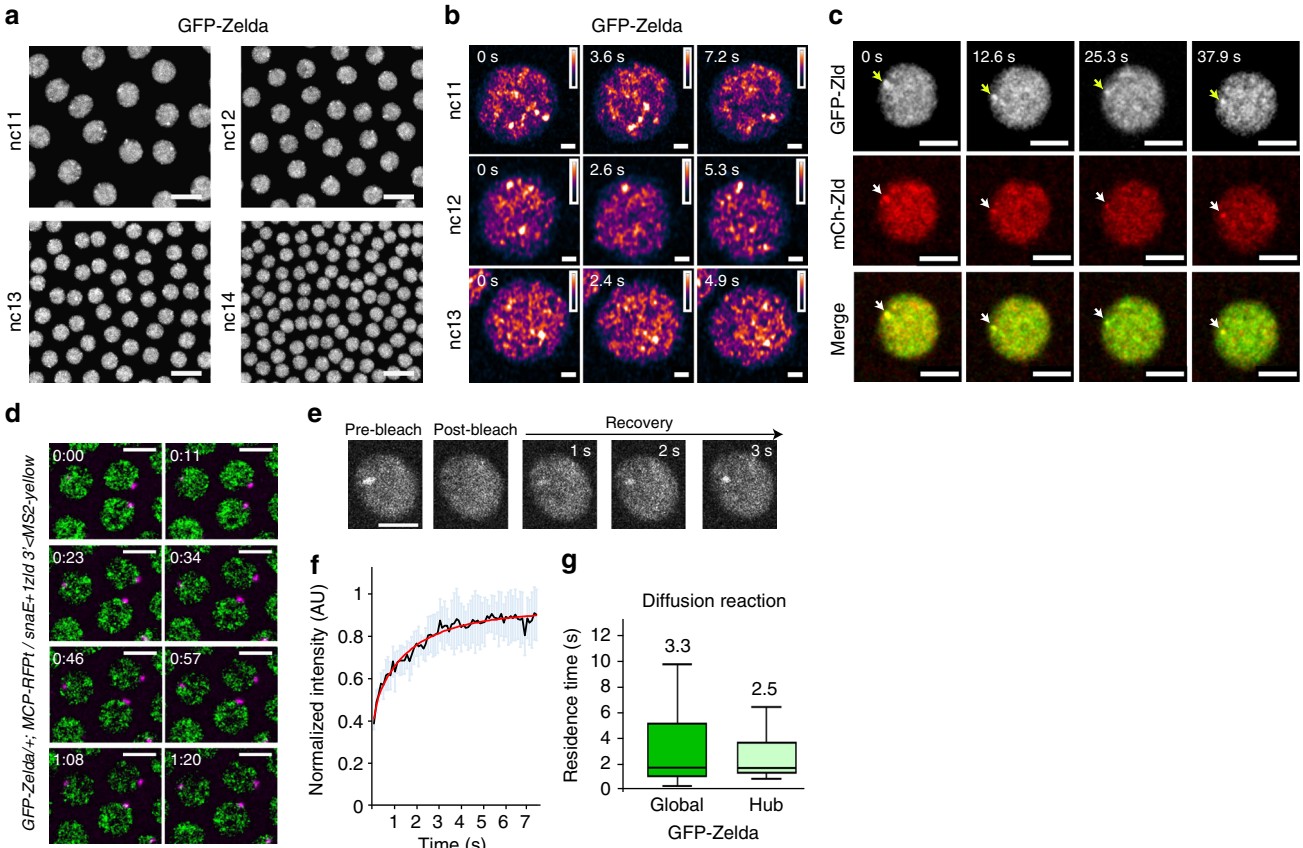

**Fig. 6** Zelda hubs kinetic properties. **a** Living *GFP-zld* embryo imaged by confocal microscopy from interphase of nc11 to early interphase of nc14. Successive representative maximum intensity projected Z-stack images showing local inhomogeneities are shown at each cycle (see also Supplementary Movie 11). Scale bars represent 10 μm. **b** Nuclei zoom from living *GFP-zld* embryos imaged by confocal microscopy from interphase of nc11 to early interphase of nc13. Successive representative images showing local hubs are shown at each cycle. Scale bars represent 1 μm. **c** Nuclei zoom from living *GFP-zld/mCherry-Zld* embryos imaged by confocal microscopy at interphase of early nc11. Successive representative images showing Zelda subnuclear hubs are shown at each cycle (an example is indicated by an arrow). Scale bars represent 5μm. **d** Living *GFP-Zld/+;MCP-RFPt/+* embryo containing the *snaE + 1Zld3'* transgene (MS2 in magenta) imaged by confocal microscopy. Successive representative maximum intensity projected Z-stacks of slices containing the MS2 signal only (see also Supplementary Movie 13). Scale bars represent 5 μm. **e** Example of a FRAP experiment on a single hub. **f** FRAP mean curve (black) and the mean of all the fits (red curve) using diffusion reaction models determined at the bleached spot for 11 nuclei from nc10–13 developing *GFP-zld* embryos. Error bars represent SD from different nuclei (light blue bars). Scale bars represent 5 μm. **g** Box plot representing estimated residence times ($1/k_{off}$) extracted with diffusion reaction model of GFP-Zld in Global (dark green) compared to Hubs (light green). Numbers above each plot represent the mean RT. Centered lines represent the median and whiskers represent min and max

consistent with the finding that Zelda participates in the *Drosophila* 3D chromatin organization during the awakening of the zygotic genome[36,37].

Using quantitative imaging approaches in living embryos combined with mathematical modeling, we have investigated a role for the pioneer-like factor Zelda in regulating mitotic memory. We demonstrated that Zelda is dispensable for transcriptional mitotic memory. While our modeling suggests that memory may be potentiated by a reduction in the number of steps required for post-mitotic transcriptional activation, we propose that Zelda accelerates this activation by decreasing the time spent at each preceding step. Our data support a model whereby Zelda binds transiently to chromatin in localized nuclear microenvironments; to potentially accelerate the timing of the transitions required prior to transcriptional activation (e.g. local chromatin organization, recruitment of transcription factors, recruitment of Pol II and general transcription factors). These dynamic properties allow the pioneer-like factor Zelda to act as a quantitative timer for fine-tuning transcriptional activation during early *Drosophila* development.

## Methods

***Drosophila* stocks and genetics**. The *yw* stock was used as a control. The germline driver *nos-Gal4:VP16* (BL4937) was recombined with an *MCP-eGFP-His2Av-mRFP* fly line (gift from T.Fukaya). RNAi were expressed following crosses between this recombinant and *UASp-shRNA-w* (BL35573) or *UASp-shRNA-zld*[4], virgin females expressing RNAi, *MCP-GFP-His2Av-mRFP* were crossed with *MS2* containing transgenic constructs males. The *GFP-zld and mCherry-zld* strains were obtained by CRISPR[25]. *GFP-ash1*[28] and *His2Av-mRFP* (BL23651) and *eGFP-bcd*[27].

**Cloning and transgenesis**. The *snaE* enhancer transgene was described in Ferraro et al[14]. The 24XMS2 tag was inserted immediately upstream of the yellow reporter gene coding sequence. We verified that our MS2 sequences do not contain any canonical binding site and we also compared synchrony of the *snaE* transgenes with our 24XMS2 to newly available MS2 loops, 24XMS2ΔZelda[38], kindly provided by Pr.Dostatni (Supplementary Fig. 7). Extra Zld-binding sites (consensus sequences CAGGTAG) were added to the *wt snaE* enhancer by PCR using the primers indicated in Supplementary Data 3. Insertion of an extra Zld-binding site in the middle of the *snaE* enhancer was done by directed mutagenesis using primers indicated in Supplementary Data 3. All transgenic MS2 flies were inserted in the same landing site (BL9750 line) using PhiC31 targeted insertion[39].

**Live imaging**. Embryos were dechorionated with tape and mounted between a hydrophobic membrane and a coverslip as described in[14]. All movies (except when specified) were acquired using a Zeiss LSM780 confocal microscope with the

following settings: GFP and RFP proteins were excited using a 488 nm and a 561 nm laser respectively. A GaAsP detector was used to detect the GFP fluorescence; a 40x oil objective and a 2.1 zoom on central ventral region of the embryo, 512 × 512 pixels, 16 bits/pixel, with monodirectional scanning and 21z stacks 0.5μm apart. Under these conditions, the time resolution is in the range of 22–25 s per frame. Two and one movie respectively for *snaE + 3Zld* in *white*-RNAi and *zld*-RNAi background were acquired on a Zeiss LSM880, keeping the time resolution is in the range of 21–22 s per frame. Image processing of LSM780 and LSM880 movies of the MS2-MCP-GFP signal were performed in a semi-automatic way using custom made software, developed in Python™. Live imaging of *GFP-zld/ +; His2Av-mRFP/ +* and *GFP-bcd/ +; His2Av-mRFP/ +* embryos were performed using a Zeiss LSM780 confocal microscope with the following settings: GFP and RFP proteins were excited using a 488 nm and a 561 nm laser respectively. A GaAsP detector was used to detect the GFP fluorescence. 512 × 512 pixels, 16 bits/pixel, images were acquired using a 40x oil objective, with bidirectional scanning and 8 stacks 1μm apart. Under these conditions, the time resolution is in the range of 5–7 s per frame. Live imaging of *GFP-zld* embryos (related to Fig. 6b) was acquired using a Zeiss LSM880 confocal microscope with the following settings: GFP protein was excited using a 488 nm laser. A GaAsP-PMT array of an Airyscan detector was used to detect the GFP fluorescence. 568 × 568 pixels, 8 bits/pixel images were acquired using a 40x oil objective, 13–14z stacks 0.5μm apart, with a 4x zoom and bi-directional scanning, with fast Airyscan in optimal mode. Under these conditions and using a piezo Z, the time resolution is in the range of 2–4 s per frame.

Live imaging of *GFP-zld/mCherry-zld* embryos (related to Fig. 6c) was acquired using a Zeiss LSM880 confocal microscope with the following settings: GFP and mCherry proteins were excited using a 488 nm and a 561 nm laser respectively. A GaAsP detector was used to detect the GFP fluorescence and a GaAsP-PMT array of an Airyscan detector was used to detect mCherry. 512 × 512 pixels, 16 bits/pixel images were acquired using a 40x oil objective, 5z stacks 1μm apart, with a 3.6x zoom and bidirectional scanning. Under these conditions, the time resolution is in the range of 12–13 s per frame. Live imaging of *GFP-zld, MCP-RFPt* (stock details available upon request) embryos containing *snaE + 1Zld3'* (related to Supplementary Movie 13) was acquired using a Zeiss LSM880 confocal microscope with the following settings: GFP and MCP-RFPt proteins were excited using a 488 nm and a 561 nm laser respectively. A GaAsP detector was used to detect the GFP fluorescence. 512 × 512 pixels, 16 bits/pixel, images were acquired using a 40x oil objective, 12z stacks 0.5μm apart, with a 4x zoom. Under these conditions, the time resolution is in the range of 11–12 s per frame. Live imaging of *GFP-zld* embryos (related to Supplementary Movie 12) was acquired using a Zeiss LSM880 using a 40x /1.3 Oil objective. Images (256 × 256 pixels), 12bits/pixel, 5z stacks 1μm apart, zoom 8x, were acquired every ≈550 ms during 100 frames. GFP was excited with an Argon laser at 488 nm and detected between 496–565 nm.

**Image analysis for transcriptional activation**. Briefly, red (nuclei) and green (MS2 RNA spots) channels were first maximum intensity projected. The analysis was divided into three parts: before mitosis (nc13), during mitosis and after mitosis (nc14). During interphases, nuclei were segmented (using mainly circularity arguments and water-shed algorithm) and tracked with a minimal distance criterion. Chromosomal splitting at the end of anaphase was automatically detected and used to define a nucleus-by-nucleus mitosis frame. During mitosis, nuclei were segmented by intensities and tracked by an overlap in consecutive frames criterion. By merging these three parts, we could give a label to each nc13 mother nucleus, their daughters in nc14 and an extra label to recognize one daughter from the other. MS2 spots were detected with a blob detection method with a user-defined threshold constant value. Detected spots were associated to the closest nuclei, inheriting their label. Finally, for each tracked nucleus, the timing of first transcriptional activation was recorded. Analyses were done with the activation timing of the two daughters for synchrony and the first active daughter for memory.

**Manual tracking**. Upon high reduction of maternal Zelda, zygotic activation is perturbed, and major developmental defects occur. Abnormal nuclear shape (example Supplementary Movie 6 and control Supplementary Movie 5) precludes their automatic segmentation. Kinetics of transcriptional activation was manually analyzed for *zld*-RNAi embryos. We implemented a spot detection algorithm to export files with the detected MS2 spot. The thresholding of the MS2 spot is consistent with all the other automatic analyses. A spatial domain was determined, taking the pseudo-furrow as a landmark and focusing on a 25μm region around it (note that activation does not seem to occur outside of this area). Nuclei in this region are visually tracked frame-by-frame to detect when activation in nc13 occurs, when mitosis occurs and to recover the first activation frame in nc14. Approximately 2 to 3-fold more nuclei from inactive mothers, compared to descendants of active mothers, were analyzed.

**Immunostaining and RNA in situ hybridization**. Embryos were dechorionated with bleach for 1–2 min and thoroughly rinsed with $H_2O$. They were fixed in fixation buffer (500 μl EGTA 0.5 M, 500 μl PBS 10x, 4 ml Formaldehyde MeOH free 10%, 5 ml Heptane) for 25 min on a shaker at 450 rpm; formaldehyde was replaced by 5 ml methanol and embryos were vortexed for 30 s. Embryos that sank to the bottom of the tube were rinsed three times with methanol. For immunostaining, embryos were rinsed with methanol and washed three times with PBT (PBS 1x 0.1% triton). Embryos were incubated on a wheel at room temperature twice for 30 min in PBT, once for 20 min in PBT 1% BSA, and at 4 °C overnight in PBT 1% BSA with primary antibodies. Embryos were rinsed three times, washed twice for 30 min in PBT, then incubated in PBT 1% BSA for 30 min, and in PBT 1% BSA with secondary antibodies for 2 h at room temperature. Embryos were rinsed three times then washed three times in PBT for 10 min. DNA staining was performed using DAPI at 0.5 μg.ml⁻¹. Primary antibody dilutions for immunostaining were mouse anti-GFP (Roche IgG1κ clones 7.1 and 13.1) 1:200; rabbit anti-GFP (Life technologies A11122) 1:100; mouse anti-Ser5-P Pol II (Covance H14, MMS-134R) 1:100; rat anti-Ser2-P Pol II (3E10) 1:5; rabbit anti-GAF (a gift from Dr.G.Cavalli) 1:250; Secondary antibodies (anti-rabbit Alexa 488-conjugated (Life technologies, A21206); anti-mouse Alexa 488-conjugated (Life technologies, A21202); anti-mouse IgM Alexa 555-conjugated (Life technologies, A21426); anti-rabbit Alexa 555-conjugated (Life technologies, A31572); were used at a dilution 1:500. Fluorescent in situ hybridization was performed as described in[14]. A Dixogygenin-MS2 probe was obtained by in vitro transcription from a bluescript plasmid containing the 24-MS2 sequences, isolated with BamH1/BglII enzymes from the original Addgene MS2 plasmid (# 31865). *Snail* probe was described in[19]. Primary and Secondary antibody for Fluorescent in situ hybridization were sheep anti-Digoxigenin (Roche 11333089001) 1/375; mouse anti-Biotin (Life technologies, 03–3700) 1/375; anti-mouse Alexa 488-conjugated (Life technologies, A21202) and anti-sheep Alexa 555-conjugated (Life technologies, A21436) 1:500. Mounting was performed in Prolong Gold. Fluorescent in situ hybridization with *yellow* probes was performed as described in[40], probe sequences will be available upon request.

**Fluorescence recovery after photobleaching**. Fluorescence recovery after photobleaching (FRAP) in embryos at nc14 (Global) was performed with the following settings: a Zeiss LSM780 using a 40x/1.3 Oil objective and a pinhole of 66 μm. Images (512 × 32 pixels), 16bits/pixel, zoom 6x, were acquired every ≈20 ms during 400 frames. GFP was excited with an Argon laser at 488 nm and detected between 507–596 nm. Laser intensity was kept as low as possible to minimize unintentional photobleaching. A circular ROI (16 × 16 pixels) 0.07 μm/pixels, was bleached using two laser pulses at maximal power during a total of ≈130 ms after 50 frames. Hub FRAP was performed in embryos from nc10 to nc13 with the following settings: FRAP was performed on a Zeiss LSM880 using a 40 × /1.3 Oil objective and a pinhole of 83 μm. Images (256 × 256 pixels), 12bits/pixel, zoom 8x, were acquired every ≈104 ms during 100 frames. GFP was excited with an Argon laser at 488 nm and detected between 496–565 nm. Laser intensity was kept as low as possible to minimize unintentional photobleaching. A circular ROI (8 × 8 pixels) 0.1 μm/pixel, was bleached using four laser pulses at maximal power during a total of ≈114 ms after 3 frames. To discard any source of fluorescence intensity fluctuation other than molecular diffusion, the measured fluorescence recovery in the bleached ROI region ($I_{bl}$) was corrected by an unbleached ROI ($I_{unbl}$) of the same nucleus and another ROI outside of the nucleus ($I_{out}$) following the simple equation:

$$I_{bl_{corr}}(t) = \frac{I_{bl}(t) - I_{out}(t)}{I_{unbl}(t) - I_{out}(t)}. \tag{1}$$

The obtained fluorescence recovery was then normalized to the mean value of fluorescence before the bleaching i.e.,:

$$I_{bl_{norm}}(t) = \frac{I_{bl_{corr}}(t)}{\frac{1}{N}\sum_{n=1}^{50} I_{bl}(n)}. \tag{2}$$

**Fitting FRAP recovery curves**. Three different analytical equations were used to fit the fluorescence recovery depending on the model of dynamics chosen.

For the pure reaction kinetics model, we started from the analytical expression of Spague et al.[41] and decline it for the case of two exchanging populations, leading to the two following equations:

one exchanging population

$$F(t) = 1 - C_{eq}e^{-k_{off}t} \tag{3}$$

two exchanging populations

$$F(t) = 1 - C_{eq_1}e^{-k_{off_1}t} - C_{eq_2}e^{-k_{off_2}t} \tag{4}$$

With $C_{eq} = k^*_{on} / (k_{off} + k^*_{on})$ as in their case.

Finally, for the reaction-diffusion model, we started from the analytical expression developed in the supplemental (Supplementary Equation 35) of[29].

$$F(t) = F_{eq}F_D(t) + C_{eq}F_{exc}(t) \tag{5}$$

with $C_{eq}$ defined as above and $F_{eq} = k_{off} / (k_{off} + k^*_{on})$. $F_D(t)$ is the fluorescence recovery due to diffusion and $F_{exc}(t)$ the fluorescence recovery due to exchange.

Since we used a Gaussian shape illumination profile, $F_D(t)$ is defined using a slightly modified version of the analytical equation of the 20th order limited development of the Axelrod model for Gaussian profile illumination and diffusion[42,43]:

$$F_D(t) = \frac{1 - e^{-K}}{K}(1 - M) + M \sum_{n=1}^{20} \frac{(-K)^n}{n!}\left(1 + n + 2n\frac{t}{\tau}\right)^{-1} \quad (6)$$

$$M = \frac{I_{(t>30\tau)} - I_0}{1 - I_0}, \quad (7)$$

where $K$ is a constant proportional to bleaching deepness, $M$ is the mobile fraction and $\tau$ is the half time of recovery. To minimize the effect of mobile fraction on $C_{eq}$, $M$ was kept between 0.9 and 1.1.

Diffusion coefficients of the different molecules were determined according to

$$D = \frac{\beta w^2}{4\tau} \quad (8)$$

with $w$ the value of the radius at $1/e^2$ of the Gaussian beam (in our case, $w = 0.56\,\mu m$ or $0.4\,\mu m$) and $\beta$ a discrete function of $K$ tabulated in ref. [44].

$F_{exc}(t)$ is defined as in ref. [29], slightly modified with respect to the Gaussian illumination, leading to the following equation:

$$F_{exc}(t) = F_\infty - \left(\frac{1 - e^{-K}}{K} - F_\infty\right)e^{-k_{off}t} \quad (9)$$

with $K$ defined as previously.

**Fluorescence correlation spectroscopy.** FCS experiments were performed on a Zeiss LSM780 microscope using a 40x/1.2 water objective. GFP was excited using the 488 nm line of an Argon laser with a pinhole of 1 airy unit. Intensity fluctuation measured for 10 s were acquired and auto-correlation functions (ACFs) generated by Zen software were loaded in the PyCorrFit program[45]. Multiple measurements per nucleus, in multiple nuclei and embryos at 20 °C were used to generate multiple ACF, used to extract kinetic parameters. The FCS measurement volume was calibrated with a Rhodamine6G solution[46] using $D_f = 414\,\mu m^2.s^{-1}$. Each time series was fitted with the reaction dominant model[29]:

$$G_D(\tau) = \frac{1}{2^{3/2}N}F_{eq}\left(1 + \frac{\tau}{\tau_{D_f}}\right)^{-1}\left(1 + \frac{\tau}{\omega^2\tau_{D_f}}\right)^{-1/2} + \frac{1}{2^{3/2}N}C_{eq}e^{-k_{off}\tau} + G_\infty, \quad (10)$$

where $F_{eq} = k_{off}/(k_{off} + k^*_{on})$, $C_{eq} = k^*_{on}/(k_{off} + k^*_{on})$, $\tau_{Df} = w^2_{xy}/4\,\mu\,D_f$ and $\omega = w_z/w_{xy}$.

For each correlogram the value of $k^*_{on}$, $k_{off}$ and $D_f$ (reaction constants and diffusion coefficient) were estimated using the PyCorrFit program implemented with Eq. 10.

Photophysics of the dye was neglected and we introduced a $G_\infty$ value to account for long time persistent correlation during the measurements. Based on the average number of Zelda molecules ($N$), connected to the $G(0)$ value following the formula $G(0) = \frac{1}{2^{3/2}N}$ and on the illumination volume, we found that Zelda concentration is around $410^{+/-90}$ nM.

**Mathematical model, general framework.** We are interested in the time needed for post-mitotic transcription (re)activation. We model this time as the sum of two variables:

$$T_a = T_0 + T_r, \quad (11)$$

where $T_0$ is a deterministic incompressible lag time, the same for all nuclei, and $T_r$ is a random variable whose value fluctuates from one nucleus to another. The decomposition in Eq. 11 can be justified by the experimental observation that all the reactivation curves (Fig. 1l) start with a nonzero length interval during which no nuclei are activated. Furthermore, $T_r$ is defined such that it takes values close to zero with nonzero probability. This property allows us to set the time origin to the instant when the first nuclei initiates transcription, in order to determine $T_r$. For large populations of nuclei, this stems to setting the time origin to $T_0$ (see Supplementary Methods).

The stochastic part of the (re)activation time is modeled using a finite state, continuous time, Markov chain.

The states of the process are $A_1, A_2, \ldots, A_{n-1}, A_n$. The states $A_i$, $1 \leq i \leq n - 1$ are metastable and OFF, i.e. not transcribing. The state $A_n$ is ON, i.e. transcribing. Each metastable state has a given lifetime, defined as the waiting time before leaving the state and going elsewhere. For the purposes of this paper, we considered that each of the states has the same lifetime denoted $\tau$. Also, the topology of the transitions is considered linear: in order to go to $A_{i+1}$ one has to visit $A_i$.

We have also tested a more complex model, with uneven lifetimes and therefore, more parameters. However, the complex model did not provide a sensibly better fit with data. Furthermore, the determination of individual

parameters of the more complex model is uncertain: many different combinations of parameter values lead to very close fit. Following Occam's razor principle, we based our analysis on the simplest, homogeneous jump model. For consistency, detailed models are described in Supplementary Methods.

**Code availability.** The image analysis software and associated graphical user interface developed for this study are available from the corresponding author upon request.

## Data availability

All relevant data supporting the key findings of this study are available within the article and its Supplementary Information files or from the corresponding author upon reasonable request.

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

## Acknowledgements

We are grateful to T.Fukaya for sharing the *MCP-eGFP-His2Av-mRFP* fly stock. We thank L.Ringrose for sharing *Ash1-GFP* fly stocks, C.Rushlow for *Zelda-RNAi* fly stocks and N. Dostatni for plasmids and eGFP-Bcd fly stocks. We thank G.Cavalli for providing the anti GAF antibody. We thank M. Bellec, L. Ciandrini, E. Bertrand, R. Feil and T. Forné for helpful discussions and critical reading of the manuscript. We acknowledge the imaging facility MRI, member of the national infrastructure France-BioImaging supported by the French National Research Agency (ANR-10-INBS-04, «Investments for the future». A.T was a recipient of a FRM fellowship and is currently sponsored by the SyncDev ERC grant. Work in the lab of M.H. was supported by NIH R01GM111694. This work was supported by the ERC SyncDev starting grant to M.L and a HFSP-CDA grant to M.L.

## Author contributions

M.L conceived the project. M.L designed the experiment with the help from J.D. M.L, J. D, J.H, C.Fz, J.L, M.D, L.M, performed experiments. M.L, J.D and O.R analyzed the data and interpreted the results. A.T developed the software for image analysis. M.D and J.D performed fly handling. C.Fd advised J.D and A.T for FRAP and FCS experiments and performed the analysis. O.R performed mathematical modeling. M.H and K.S generated the *GFP-zld and mCherry-zld* lines and commented the data. M.L wrote the manuscript with help from J.D and O.R. All authors discussed, approved and reviewed the manuscript.

## Additional information

**Competing interests:** The authors declare no competing interests.

