## [Peer Review File · Nature Communications]

Reviewer #1 (Remarks to the Author):

The manuscript comprises a quantitative real time analysis in living *Drosophila* embryos of the kinetics of transcriptional activation and of chromatin binding by the pioneer transcription factor Zelda. There are three main claims of the paper:

1) The authors show using a reporter gene that Zelda binding sites quantitatively affect the timing of transcriptional activation. Both the position of added Zelda binding sites in relation to the promoter, and the number of added binding sites have an effect, with more or better positioned sites generally leading to an accelerated transcriptional activation of the reporter. The authors show that this accelerated activation is manifested in a coordinated manner among different nuclei and conclude that Zelda impacts on synchrony of activation (page 3 line 83).

2) The authors use the same live imaging set up in combination with mathematical modelling to investigate whether Zelda works as a mitotic bookmarking factor. The experimental evidence points strongly away from a role in mitotic bookmarking, which the authors report as a surprising conclusion. The modelling is used to further investigate a potential role for Zelda in bookmarking, but I found the authors' conclusions regarding what they learned from the modelling about Zelda as a bookmarking factor to be somewhat unclear (page 5 line 161 - 170; see specific comments below).

3) The authors investigate kinetic properties of Zelda chromatin binding using Zelda-GFP in combination with time lapse imaging, FRAP and FCS. Together these data show that Zelda does not detectably bind mitotic chromatin, and that it binds rapidly (after mitosis) and dynamically to interphase chromatin. The authors also show that interphase binding is non-homogeneous, and propose that local high concentrations of Zelda may compensate for the observed transient binding seen on a global level in FCS experiments (page 1 line 35 – line 37).

There are some very nice experiments here. The experiments are generally performed to a high technical standard, and the imaging of transcriptional activation for different reporters is a terrific tool with which to follow activation kinetics in real time. However, the main claim to novelty, and the most convincing evidence, is that Zelda is not a bookmarking factor. The authors do propose alternatives, but the claims for a role for Zelda in (1) synchrony, (2) shortening the length of pre-initiation transitions, or (3) acting at local high concentrations, are based on indirect evidence and are less convincing. In the absence of a clear mechanistic insight into how Zelda does function, the paper in its current form reads as a collection of different good experiments, but without a clear conceptual advance beyond the lack of bookmarking function.

Incidentally, the evaluation of novelty is severely hindered by the complete lack of references in the introduction. The section is labelled “abstract/intro” in the paper, but due to its length it does not fulfil the criteria of the former, and due to the lack of citations it cannot be considered as the latter. This should be corrected upon any revision of the manuscript.

I give specific comments below relating to each of the three parts indicated above.

1) Zelda and transcriptional activation.

a) Fig 1A is very hard to read- the diagram should be bigger.

b) Fig 1E' I cannot understand from the legend, the text, or the methods, what is automatically segmented here. What are the different colours?

c) Images with dots (F, and F'). The dots are very hard to see if not blown up on screen. It might be advisable to show a zoom of a smaller part.

d) Fig 1H-L define “nc” (I presume nuclear division cycle?). I guess this image is in the expression domain, but it is not defined where in the embryo these shots were located. Show diagram or indicate on B-D where these images come from and where “50um from the furrow” is located.

e) Page 4 line 107. “Adding zld binding sites boosted the spatio-temporal response to the dorso-ventral gradient”. I am not sure what this means. Looking at the figure (S1C-D) it shows that the reporters with extra Zelda sites get activated faster, but I don't see a spatial component there. Indeed I found this part difficult to follow because it assumes a knowledge of the Dorsal spatial gradient. This should be defined in the text or indicated in a figure.

f) The claim to a role for Zelda in synchrony is based on % active nuclei within a defined spatial pattern (Page 3 line 88). I had to look very carefully at figure legends to find a definition of this “defined spatial pattern”. It should be explained in the main text. The authors do not consider the potential explanation that the pattern simply depends on synchrony of the upstream activators of the snail enhancer (e.g. Dorsal and twist). Can the authors exclude this? Is Zelda really introducing synchrony or only thresholding an existing synchrony?

2) Bookmarking and modelling.

I found the description and conclusions of the model difficult to follow. Some specific recommendations:

a) (Fig 2E, F) and page 5 line 146. Delays of transcription are defined as “waiting times prior to the first detected initiation event”- but what is time 0? In the model description (page 17 line 551)

“post – mitotic reactivation” is mentioned. Finally on line 584 (methods) I found the statement: “The time origin was set at the end of mitosis” – but no information on how this time point is measured experimentally. This should be made clear in the main text description of the model. It is essential for understanding the model and the comparison to data, that this time point is explained precisely in the main text at the first opportunity. It is also not clear from the main text or anywhere in the manuscript, whether the waiting time measurements span only one or several cell divisions (given the time scales in Figure 2E I presume they are within the interphase of cycle 14, but the reader has to have some background knowledge and do some calculations to work this out.

b) Line 156 “Fitting our data revealed that ...” – the model is fitted to the data and not vice versa.

c) Fig 2E: show parameter a (number of transitions) and parameter b (length of each) on the diagram. Indicate here and in the main text, that b is assumed the same for each transition.

d) Page 5 Line 159 “This estimated memory time is sufficiently large to guarantee transmission of transcriptional state across mitosis for both types of mothers”. This statement (I presume relating to the estimate for parameter b) is extremely difficult to understand for several reasons. Firstly, this is the first time parameter b has been referred to as “memory time”. Secondly, the current description of the model appears to relate to post-mitotic activation within interphase of cycle 14, which contains several transitions of length b, so it does not encompass mitosis (as far as I can work out from the description). Finally, without any statement of the length of mitosis, this statement is anyway uninterpretable.

e) Fitting the model to data on different Zelda transgenes appears to result in shorter “b” estimates (i.e. faster transitions). This data should be shown in the main Figure in the same format as Figure 2E, and explained and discussed more thoroughly (see f below).

f) Page 5 line 166-170. The conclusion of the modelling section departs completely from Zelda and the reader is left rather in the dark. What do we learn about Zelda and bookmarking from the modelling that we could not have learned without it? The model is a very nice way to dissect potential explanations for different results, but it is not explained clearly enough for the reader to appreciate this, and in its current form the paper runs the risk of increasing readers mistrust of modelling rather than enlightening them to its power.

g) The conclusions of the model are oversold in the discussion. The authors state (page 9 line 276) “Zelda accelerates (post mitotic) activation by decreasing the time spent at each preceding step.” The authors should acknowledge that the model is phenomenological rather than mechanistic. The existence of transitions is an integral component of the model, and not based on any knowledge of whether these transitions exist, and if so, what they are in mechanistic terms. Differences in the length and number of transitions nicely result from fitting a model to data, but models with other structures (e.g., in which activation does not proceed via discrete steps) could give other predictions.

h) Table S2: Units for b are not given. I assume seconds?

i) In summary, the model deserves better explanation and more space for its merits to become apparent.

3) Live imaging and kinetic analysis of Zelda chromatin binding.

a) Page 8 line 246. The authors compare Zelda kinetics to those of ASH1, stating that ASH1 is a mitotic bookmarking factor. This is incorrect: ASH1 has been shown in the paper cited (ref 26) to be substantially retained on chromatin throughout mitosis, but has not to date been shown to play a role in bookmarking. The fact that it binds mitotic chromatin means it is a strong candidate for a bookmarker, but a role in accelerated gene activation of previously activated genes has not been demonstrated. Curiously the authors do not mention the fact that ASH1 binds mitotic chromatin, instead comparing its kinetic binding properties during interphase to those of Zelda. The comparison of interphase binding of ASH1 and Zelda is irrelevant to the bookmarking discussion. The most relevant feature of binding that might indicate a potential role in bookmarking for Zelda is mitotic binding: which the authors do not detect.

b) Notwithstanding the discussion of relevance of interphase kinetics, the FCS data should be presented correctly and put in their correct context. The authors measure somewhat different values to those reported in Steffen et al (2013) for k^*_{on} and k_{off} from FCS for ASH1 GFP but do not mention this. I presume the transgenic flies are the same as those used in Steffen et al (page 13 line 412 gives the reference but does not state whether the authors obtained the flies directly from Steffen or if they were generated newly. The lack of acknowledgment rather suggests the latter). A comparison and potential reasons for the discrepancy should be presented (e.g, potentially the confocal volume for FCS was calculated differently, or the cycle number at which kinetics were determined was different?).

c) The authors do not correctly interpret the meaning of k^*_{on} . k^*_{on} extracted from FRAP or FCS data is defined as $k_{on} \times [X]$, where $[X]$ is the concentration of free binding sites at equilibrium. As $[X]$ is unknown and is different for different proteins, it is meaningless to compare k^*_{on} for Ash1 and Zelda. Furthermore k_{on} itself is also concentration dependent - in the absence of information on absolute Zelda concentrations, comparison of Zelda binding rates to its dissociation rates is meaningless (Page 8 line 254). In contrast, k_{off} is independent of binding site number and protein concentration; $1/k_{off}$ (residence time) is thus a useful measure. The conclusion that Zelda binding is (on a global level) dynamic, is valid.

d) The last sentence of the discussion (page 9 line 284): "In this case, we propose that mitotic bookmarking factors prevent the decline to profound states during mitosis." would benefit from rewording to improve clarity ("profound states" is unclear) and focus (is the paper about bookmarking or Zelda?)

Reviewer #2 (Remarks to the Author):

Dufourt et al. presented an in-depth study on the mechanisms of Zelda (Zld) in promoting enhancer priming in the early embryo (nc11 to nc14), combining in vivo imaging of transcription kinetics, mathematical modeling of the kinetics, and FRAP/FCS to measure Zld binding dynamics. Their observations that Zld is highly mobile inside the nucleus and interacts with DNA transiently are in concordance with recent works showing that transcription factors in complex eukaryotes bind transiently but repeatedly to specific nuclear locations. It is of interest that they found that Zld distributions inside nuclei, both in live embryos and fixed immunofluorescence stained embryos, are heterogeneous, echoing several recent works looking at other transcription factors such as Bcd and Ubx. Their unexpected finding that Zld alone is insufficient for mitotic memory (hysteresis) is a highlight of this study, demonstrating the need to address mechanistic questions through direct observation in vivo. Their conclusion that Zld accelerates the rates for all steps leading to transcription is supported through empirical observations and modeling.

However, the authors should address the following major issues:

1. The authors designed different constructs based on the *snaE* enhancer with Zld sites inserted at 5' or 3' end of the enhancer with respect to the promoter and with different numbers of Zld sites. A recent publication (Crocker, Tsai, & Stern. 2017, Cell Rep. 18:287-296.) also explored the role of Zld binding sites in transcriptional activation, observing that increasing Zld sites led to more nuclei showing gene expression but did not change the output levels of gene expression in active nuclei. The authors should discuss the similarities and differences of their findings in relationship to the work published in Cell Reports.
2. Have the authors observed the effect of placing a Zld site in the middle of the enhancer (e.g. as is the case in the Cell Reports paper)? If so, what is its effect on the kinetics of transcription? Is it in between the 5' and the 3' version? Can the authors discuss the impact on Zld site placement (topology) on their ability to speed up transcriptional activation?
3. For the modeling work, as shown in Fig. 2E & F and summarized in Table S1, the variations in the a moments (number of transitions) between active and inactive nuclei, specifically between 1Zld 3', 2Zld, and 3Zld, present a less consistent picture compared to the b moments. Some configurations of Zld sites apparently led to near 2-fold changes in the values of a with previously inactive nuclei whereas some configurations of Zld sites yielded larger values of a for active nuclei than inactive nuclei. Adding Zld sites actually led to increases in a more often than not. The apparent interpretation is that some configuration of Zld sites could bypass up to 1 or 2 steps needed for transcriptional activation in inactive nuclei but work against nuclei previously bookmarked through other mechanisms. This means that while Zld may not be fully responsible for the mitotic memory observed at for this particular locus with the *snaE* enhancer, it could still function beyond simply accelerating the kinetics of the individual steps. Alternatively, is this a limitation of the model chosen or are the errors in the fits for a large?

4. Have the authors applied the mathematical modeling shown in Fig. 2E and F to their RNAi Zelda experiments to see if Zld depletion leads to a consistent increase in the time spent in each step (the parameter b)?

5. In the FCS analysis, the authors found that Zld binding inside the nucleus is transient and not very stable. However, the authors also noted in the methods that their fitting model does not take long time correlation into consideration. In their live imaging of GFP-Zld (Figure 4H-J), they observed stable loci that contained long-lasting Zld signals (e.g. the blue trace in Figure 4I). The 0.35 s dwell time from the FCS experiments contrasts with their live imaging experiments with loci continuously showing Zld signal over 30+ s. The author should clearly address the possible reasons for this discrepancy. Are the stable loci areas where Zld constantly bind to, showing high levels of activity even though individual binding events are short and unstable? Is it because that their FCS fitting explicitly exclude events longer than 1 s, thereby not including any more stable interactions in their analysis?

Minor issues in the manuscript:

1. Line 255: The authors refer to a Figure S5D. Should this be Figure S4D?

2. Line 519-520: The authors describe a 40x water objective with an NA of 1.4. The refractive index of water is only 1.33. Is this a mistake?

3. Figure 4C: The authors should show the result of their fit with the raw FCS to give a sense of the quality of the fit.

Overall, I find the experiments presented in this work well-designed and the data gathered of high quality. The mathematical modeling of the data provides intriguing insights into the possible mechanisms and dynamics of Zld, challenging the conventional view of pioneering factors and stressing the dynamic nature of transcription in vivo. I would encourage the authors to consider and discuss alternative interpretations (even if they end up discounting them after consideration) on the functions of Zld. Specifically, their data suggests that Zld may function more than simply accelerating the kinetic rates of the steps leading to transcriptional activation, even if mitotic memory in the *snfE* enhancer comes from an alternative mechanism. If the authors can address the major questions raised, I would recommend this work to be published.

Reviewer #3 (Remarks to the Author):

Dufourt et al. address important questions about the effect of the transcription activator Zelda on transcriptional memory in the early *Drosophila* embryo, using elegant imaging and image analysis

methods. They make use of synthetic reporters (containing the *sna* promoter and a truncated version of its shadow enhancer with different numbers of Zelda binding sites) to show that 1) Zelda accelerates transcriptional activation but 2) it does not enhance transcriptional memory. In addition, the dynamics of Zelda is investigated, and shown to be consistent with transient binding to chromatin, leading the authors to conclude that 3) Zelda dynamics is not consistent with that expected of a pioneer factor and to propose that its “low binding rates” to chromatin could be compensated by accumulation of Zelda in high concentration nuclear microdomains.

Part of the results presented are novel (effect of Zelda on transcriptional memory, dynamics of Zelda inside nuclei), and represent a very interesting contribution to the field. I have, however, reservations concerning both the modelling of the transcription process and the part of the work dealing with Zelda's dynamics.

General comments:

1. The influence of Zelda on transcriptional memory is the most convincing and novel aspect of the paper. Why then have a title that only mentions role of Zelda as a temporal coordinator of gene activation (something which has already been highlighted in several other papers)?

2. The importance of looking into the dynamics of the binding between Zelda and DNA, and its relevance to the discussion of whether Zelda is a pioneer factor, is not well articulated in the abstract/introduction.

3. Since transcriptional memory seems very weak in the presence of sufficient Zelda binding sites (and for the intact shadow enhancer), how is memory relevant for embryo development?

4. It would be useful to show the data obtained with the intact enhancer in Fig. 1 (schematic representation and number of Zelda binding sites in Fig. 1A, and synchrony curve in Fig. 1M).

Modeling:

5. The conclusions drawn from the analysis of the distributions of waiting are weakened by the fact that only one model is considered (based on the assumption that all states have the same average

lifetime). This model should be compared to other equally plausible models (e.g. models with a fixed number of states each with a different lifetime).

6. Line 152: “nuclei require at most three transitions to become active”. Should it be “rate-limiting” transitions (as the distribution of waiting time would likely only reflect those transitions that are rate-limiting)?

7. Lines 156-159: It seems a bit of a misrepresentation to write that the main difference between nuclei descending from active vs. inactive mothers is seen for parameter b , which passes from 3.05 to 2.04, a ~30% decrease, while parameter a passes from 208 s to 147 s (also a ~30% decrease).

8. In Fig 2F, it would be good to also schematize the effect of Zelda on the two populations of nuclei.

Dynamic properties of Zelda:

9. The authors mention (line 241) that at the end of mitosis Zelda “comes back to the nucleus very rapidly”. What does that mean, and how does that compare with how quickly other factors are imported in nuclei after mitosis (e.g. Bicoid, as studied by Gregor et al., 2007)?

10. How come the FRAP and FCS experiments give such different values of the diffusion coefficients?

11. FRAP experiments: The photobleaching duration should be compared to the recovery time (if both photobleaching time and recovery time are of the same order of magnitude, the FRAP experiment does not capture the true dynamics of the fluorophore, due to the “halo effect”). From the methods, it is unclear what the duration of that photobleaching period was, 130 ms or 260 ms (given that two laser pulses are mentioned)? If it was 260 ms, as the recovery time was ~ 0.5 s (Fig. 4B), the halo effect may be artificially decreasing the measured diffusion coefficient.

12. FCS experiments: The autocorrelation functions shown both for GFP-Zelda (Fig. 4C) and GFP-Ash1 (Fig. S4C) do not fully decay to 0 at the longest times shown. This is a sign of the presence of dynamic processes with characteristic times on the same order of magnitude or larger than the measurement time (10s), and calls into question the validity of the equilibrium model used to fit the

data (Eq. 6). The authors mention that the data above 1 s was not analyzed, but this does not eliminate the issue.

13. Since photobleaching is a concern in FCS experiments in live organisms, the authors should report on the excitation intensity, or, at least, on the average number of photons collected per Zelda-GFP.

13. Can the authors use their FCS data to estimate the concentration of Zelda?

14. The part of the manuscript dealing with the inhomogeneous distribution of Zelda in nuclei seems quite preliminary and almost an after-thought. To be useful, this distribution should be characterized, not just mentioned.

15. An inhomogeneous distribution also raises question as to how the FRAP and FCS experiments were conducted: where they performed in a region of high or low Zelda concentration?

Minor comments:

16. The sentence "This emphasizes the concept that synchrony is distinct from that of memory" (line 198) is unclear. Maybe because it is grammatically incorrect (what does "that" refer to?). Or maybe because the concept of synchrony has not been well defined. If synchrony means that nuclei all start expressing at the same time (meaning both nuclei from active and inactive mother), then doesn't it immediately imply that memory is irrelevant?

17. What does "zoomed 6x" (line 492), "4x zoom" (line 446) and "2.1 zoom" (line 429) mean?

18. Fig. 4C: The fit to the FCS data should be shown.

19. line 397: Should be "The circled cluster in panel (H)" (not (J))".

20. line 520: A water objective should not have a NA above ~ 1.3 .

21. line 527: What is “mu” in the definition of τ_{Df} ?

22. line 560: The lifetime is the average waiting time.

23. Some references are not properly formatted (e.g. ref 24, ref. 27)

We are very grateful to the reviewers for their various comments that were extremely helpful to improve our manuscript. We did our best to incorporate the majority of these excellent suggestions. Below we provide a detailed, point-by-point account of the changes in the revised manuscript.

The main text now includes a full introductory section, a completely rephrased modeling section and a new paragraph discussing Zelda hubs and Zelda binding kinetics within these hubs.

Figures have been modified accordingly, with a new Figure devoted to modeling (now Fig. 4), a new Fig. 6 (hubs) and 3 new Supplementary Figures, Supplementary Fig. 4 dedicated to modeling, Supplementary Fig. 6 describing FRAP results and Supplementary Fig. 7 as a control with new MS2 stem-loops. Indeed during the revision process, the Dostatni's lab reported that older versions of MS2 loops (e.g Lucas et al CB 2013) contained cryptic Zelda sites, possibly responsible for artifactual activations (Lucas et al., Biorxiv 2018). Even if our MS2 sequences is a newer version, different from that used in Lucas et al., 2013, in order to have an extra control line, we created a new transgenic line with the newly designed MS2 sequence by Dostatni's lab. We found very similar kinetics, shown in Supplementary Fig. 7.

The revised manuscript is supported with 8 new movies, listed in the Supplementary Methods and 2 Supplementary Tables depicting parameters estimated using our model and using an alternative model suggested by the reviewers.

We also provide a Figure to answer to Reviewer 2, point II.1, accompanying this rebuttal letter.

Reviewers' comments:

Reviewer #1 (I):

We are grateful to Reviewer 1 for his/her detailed and constructive comments.

Reviewer 1 was concerned about the novelty of our study. Following his/her advice we have now included a full introductory section that ends with a summary of our new findings.

In summary, we think that the effect of Zelda on temporal dynamics (synchrony and memory) and its action in local hubs constitute the main novel aspects of this study.

I.1) Zelda and transcriptional activation.

I.1.a) Fig 1A is very hard to read- the diagram should be bigger.

The size of the diagram was increased in Fig. 1a.

I.1.b) Fig 1E' I cannot understand from the legend, the text, or the methods, what is automatically segmented here. What are the different colours?

The tracked nuclei are given a random color; this is now explained in the legend of Fig. 1. '(e) and associated automatic segmentation, *where each tracked nuclei is given a random color (e)*' page 24.

I.1.c) Images with dots (F, and F'). The dots are very hard to see if not blown up on screen. It might be advisable to show a zoom of a smaller part.

As suggested by the Referee, we added a zoom as insets to Fig. 1 panel f and f'.

I.1.d) Fig 1H-L define “nc” (I presume nuclear division cycle?). I guess this image is in the expression domain, but it is not defined where in the embryo these shots were located. Show diagram or indicate on B-D where these images come from and where “50um from the furrow” is located.

Definition of nc (nuclear cycle) is now introduced in the text (page 3). *‘Both enhancers are bound by Zelda (Zld) at early nuclear cycles (nc)’*.

We have now added a new panel to Supplementary Fig. 1a depicting a typical imaged area (for MS2 type of imaging) within an entire embryo.

I.1.e) Page 4 line 107. “Adding zld binding sites boosted the spatio-temporal response to the dorso-ventral gradient”. I am not sure what this means. Looking at the figure (S1C-D) it shows that the reporters with extra Zelda sites get activated faster, but I don't see a spatial component there. Indeed, I found this part difficult to follow because it assumes a knowledge of the Dorsal spatial gradient. This should be defined in the text or indicated in a figure.

We agree that our description on this result was too succinct. This was done on purpose since Zelda potentiation of other TF binding was published with other approaches (cited in the text). We thank the reviewer for pointing out the lack of clarity of Supplementary Fig. 1c-d. We have now modified Supplementary Fig. 1d to clearly show the effect of extra-Zelda sites on the timing of activation along the dorso-ventral axis.

I.1.f) The claim to a role for Zelda in synchrony is based on % active nuclei within a defined spatial pattern (Page 3 line 88). I had to look very carefully at figure legends to find a definition of this “defined spatial pattern”. It should be explained in the main text. The authors do not consider the potential explanation that the pattern simply depends on synchrony of the upstream activators of the snail enhancer (e.g. Dorsal and twist). Can the authors exclude this? Is Zelda really introducing synchrony or only thresholding an existing synchrony?

We understand that this point was raised, however please note that we explained our definition of a ‘spatially defined pattern’ in a full paragraph (page 4) and illustrated it with Fig. 1g,g’. To make our definition clearer, we have now added *‘... among a spatially defined pattern (i.e the presumptive mesoderm, 50µm around the furrow)’* page 4.

The referee points to the possible effect of upstream activators on the synchrony of *sna* expression. At the scale of the whole embryo, we completely agree that this could indeed occur. However, in the restricted area that we quantify (*i.e* 50um around the furrow, now illustrated by Supplementary Fig. 1a), we know that the main activators, Twist and Dorsal are homogeneously distributed and non-limiting. Thus we exclude that the observed effects are due to differential synchrony of Twist or Dorsal.

I.2) Bookmarking and modelling.

I found the description and conclusions of the model difficult to follow. Some specific

recommendations:

As suggested by all reviewers, we rewrote the part of the manuscript describing our mathematical model and its conclusions. We also devoted dedicated new figures (now Fig. 4, Supplementary Fig. 4, Supplementary Table 1 and 2) depicting the estimated parameters and a Supplementary Methods section with more details.

I.2.a) (Fig 2E, F) and page 5 line 146. Delays of transcription are defined as “waiting times prior to the first detected initiation event”- but what is time 0? In the model the statement: “The time origin was set at the end of mitosis” – but no information on how this time point is measured experimentally. This should be made clear in the main text description (page 17 line 551) “post – mitotic reactivation” is mentioned. Finally on line 584 (methods) I found of the model. It is essential for understanding the model and the comparison to data, that this time point is explained precisely in the main text at the first opportunity.

We thank the reviewer for pointing out this confusion. We have now detailed the meaning of this T0 in the Figure Legend of Fig. 4b page 26 *‘(b) Cumulative distribution functions of the random time to transcription in nc14. The origin of time is the end of mitosis nc13 to nc14, (determined by our automatic software, see methods) added to the time required to detect the first activation (proper to each genotype)’*. The need for T0 is justified by the shape of the experimental reactivation curves, which is now explained in the Methods part (page 22, equation11).

It is also not clear from the main text or anywhere in the manuscript, whether the waiting time measurements span only one or several cell divisions (given the time scales in Figure 2E I presume they are within the interphase of cycle 14, but the reader has to have some background knowledge and do some calculations to work this out.

The timing of activation is indeed measured in nuclear cycle 14. This is now indicated in Figure legend of Fig. 1-4.

I.2.b) Line 156 “Fitting our data revealed that ...” – the model is fitted to the data and not vice versa.

We thank the reviewer for indicating this to us this mis-formulation. We rephrased this part, page 8. *‘Taking advantage of the significant number of nuclei tracked in this study, we could fit this model to our data and estimate more accurately the parameters ‘a’ and ‘b’*”.

I.2.c) Fig 2E: show parameter a (number of transitions) and parameter b (length of each) on the diagram. Indicate here and in the main text, that b is assumed the same for each transition.

The parameters ‘a’ and ‘b’ are now indicated in Fig. 4 and in Supplementary Table 1. Furthermore, as nicely proposed by Reviewer 2 (II.3) and by Reviewer 3 (III.5) we now

provide two alternative models: 1-as in the original version one model consists of constant transition timings ('b' is constant, now clarified in the main text page 8). 2-a new alternative model (described in the Supplementary Methods and in Supplementary Table 2), where 'b' is not assumed constant. However, as discussed in the methods section page 23, when fitted to our data, this model does not increase the goodness of fit and predicts parameters with low confidence.

I.2.d) Page 5 Line 159 "This estimated memory time is sufficiently large to guarantee transmission of transcriptional state across mitosis for both types of mothers". This statement (I presume relating to the estimate for parameter b) is extremely difficult to understand for several reasons. Firstly, this is the first time parameter b has been referred to as "memory time". Secondly, the current description of the model appears to relate to post-mitotic activation within interphase of cycle 14, which contains several transitions of length b, so it does not encompass mitosis (as far as I can work out from the description). Finally, without any statement of the length of mitosis, this statement is anyway uninterpretable.

We thank the referee for this point and agree that the description of the model could have been made clearer. By devoting an entire main Figure (Fig. 4) and a new supplementary Figure (Supplementary Fig. 4) to the modeling part, we could rephrase our explanations (pages 8 and 9) with hopefully a clearer message. As correctly noticed by the referee, the model was fitted to post-mitotic data and concerned post-mitotic transitions only. However, under some hypotheses, we can also encompass mitosis. Considering that backward transitions can occur during mitosis, the advantage in terms of number of jumps to activation of descendants from active mothers is reduced during mitosis. Simulations of our model under such hypothesis predict that the ratio $a_{\text{inactive}} / a_{\text{active}}$, a measure of memory, increases with $(b_{\text{active}} + b_{\text{inactive}})/2$, a measure of transition timings, and decreases with mitosis length (estimated at 5 minutes for nc13/nc14). This prediction agrees with our data and provides a more nuanced and quantitatively rigorous statement of what we meant. The full description of the model is now provided in the Supplementary methods and the corresponding result is illustrated in Fig. 4e and discussed page 9.

I.2.e) Fitting the model to data on different Zelda transgenes appears to result in shorter "b" estimates (i.e. faster transitions). This data should be shown in the main Figure in the same format as Figure 2E, and explained and discussed more thoroughly (see f below).

The estimates are now provided in a dedicated figure, now Fig. 4 and explained thoroughly in the main text.

I.2.f) Page 5 line 166-170. The conclusion of the modelling section departs completely from Zelda and the reader is left rather in the dark. What do we learn about Zelda and bookmarking from the modelling that we could not have learned without it? The model is a very nice way to dissect potential explanations for different results, but it is not explained clearly enough for the reader to appreciate this, and in its current form the paper runs the risk of increasing readers mistrust of modelling rather than enlightening them to its power.

We understand why this point was raised and agree that the original version of the manuscript was confusing.

By extending our modeling to mitosis (Fig. 4e), we could rationalize the relationship between memory and transitions lifetimes. Our conclusion is that Zelda-mediated acceleration of reactivation dynamics has a negative impact on memory.

Furthermore, as suggested by Reviewer 2, we have now examined the predictions of our model in the case of *Zelda-RNAi* data. The effect on parameter 'b' is very clear ('b' values increase upon Zelda depletion), which strengthens the prediction that Zelda speeds transitions. The *Zelda RNAi* genotype also leads to large $a_{\text{inactive}}/a_{\text{active}}$ ratio, reflecting a strengthened memory bias.

Taken together, these results now allow drawing better conclusions, discussed in the main text (Modeling section, pages 8 and 9).

I.2.g) The conclusions of the model are oversold in the discussion. The authors state (page 9 line 276) "Zelda accelerates (post mitotic) activation by decreasing the time spent at each preceding step." The authors should acknowledge that the model is phenomenological rather than mechanistic. The existence of transitions is an integral component of the model, and not based on any knowledge of whether these transitions exist, and if so, what they are in mechanistic terms. Differences in the length and number of transitions nicely result from fitting a model to data, but models with other structures (e.g., in which activation does not proceed via discrete steps) could give other predictions.

We totally agree that our model is phenomenological and that, at this stage, we cannot dissect the mechanistic/biochemical nature of each transition of the model. We have added '*Although purely phenomenological, this model suggests that transcriptional memory is supported by a sequence of..*' (pages 9 and 10) to the conclusion of the modeling section.

However, phenomenology is a good start for more refined mechanistic modeling. In our case, the mathematical model provides an interesting framework to make testable predictions.

For instance, the correlation between memory bias and transition lifetimes is a highly non-trivial prediction of the model, that is validated by the data. We have shown and tested experimentally that, in conditions where the 'b' parameter is small (fast transitions), the difference between descendants of active and those of inactive mothers is reduced.

Another interesting model prediction is the number of transitions. Starting with no a priori limit for this number, we found that three transitions (3 nonproductive states, OFF1, OFF2 and OFF3) are enough to explain our data. Importantly, we generated transcriptional temporal traces in hundreds of nuclei, thereby obtaining large datasets that can be fitted with high confidence. We strongly believe that the qualitative model predictions, combined with the numerical data resulting from fitting the models to the data, are meaningful *per se* and will prove to be useful for future studies.

I.2.h) Table S2: Units for b are not given. I assume seconds?

Yes indeed, 'b' parameter is given in seconds and this is now indicated in the text and in Fig. 4d, '*parameter 'b' (expressed in seconds, considered the same for all transitions)*' (page 8).

I.2.i) In summary, the model deserves better explanation and more space for its merits to become apparent.

We fully agree and now provide extensive explanations and Figure/Tables (Fig. 4, Supplementary Fig. 4, Supplementary Table 1 and 2).

I.3) Live imaging and kinetic analysis of Zelda chromatin binding.

I.3.a) Page 8 line 246. The authors compare Zelda kinetics to those of ASH1, stating that ASH1 is a mitotic bookmarking factor. This is incorrect: ASH1 has been shown in the paper cited (ref 26) to be substantially retained on chromatin throughout mitosis, but has not to date been shown to play a role in bookmarking. The fact that it binds mitotic chromatin means it is a strong candidate for a bookmarker, but a role in accelerated gene activation of previously activated genes has not been demonstrated. Curiously the authors do not mention the fact that ASH1 binds mitotic chromatin, instead comparing its kinetic binding properties during interphase to those of Zelda. The comparison of interphase binding of ASH1 and Zelda is irrelevant to the bookmarking discussion. The most relevant feature of binding that might indicate a potential role in bookmarking for Zelda is mitotic binding: which the authors do not detect.

We are grateful to Reviewer1 for pointing out this inaccuracy. Indeed, so far there are no published data clearly showing that Ash1 acts as a mitotic bookmarker in *Drosophila*. We rephrased this sentence on page 11: '*FCS experiments for Zelda were compared to Ash1, whose kinetics have been documented by both FRAP and FCS in early Drosophila embryos*²⁸'.

I.3.b) Notwithstanding the discussion of relevance of interphase kinetics, the FCS data should be presented correctly and put in their correct context. The authors measure somewhat different values to those reported in Steffen et al (2013) for k^*_{on} and k_{off} from FCS for ASH1 GFP but do not mention this. I presume the transgenic flies are the same as those used in Steffen et al (page 13 line 412 gives the reference but does not state whether the authors obtained the flies directly from Steffen or if they were generated newly. The lack of acknowledgment rather suggests the latter). A comparison and potential reasons for the discrepancy should be presented (e.g, potentially the confocal volume for FCS was calculated differently, or the cycle number at which kinetics were determined was different?).

The FCS experiments were performed on the same Ash1-GFP stock as that of the publication Steffen et al. We apologize for this unfortunate oversight. We have now acknowledged the lab of Dr L.Ringrose for sharing this stock.

From our FCS experiments, we estimate a diffusion coefficient for Ash1 around $4.5 \mu\text{m}^2.\text{s}^{-1}$. Steffen et al., report a very similar D_i , $D_i=4.98 \mu\text{m}^2.\text{s}^{-1}$. Thus our estimates are very similar.

As pointed out by the referee, in our original submission, our estimate for Ash1 k_{off} were slightly different from that reported by Steffen et al, k_{off} (Ash1) $\sim 5\text{s}^{-1}$. In the revised manuscript, based on the very insightful Reviewer comment, we removed all FCS curves subject to photobleaching by direct analysis of each fluorescence fluctuation used to

generate the correlogram. Examples of such correlograms are given for Zelda-FCS (Fig. 5d') and for Ash1-FCS (Supplementary Fig. 5e'). With this new filtering, we estimate a mean k_{off} for Ash1 of $4.6s^{-1}$, in complete agreement with Steffen et al results.

c) The authors do not correctly interpret the meaning of k^*_{on} . k^*_{on} extracted from FRAP or FCS data is defined as $k_{on} \times [X]$, where $[X]$ is the concentration of free binding sites at equilibrium. As $[X]$ is unknown and is different for different proteins, it is meaningless to compare k^*_{on} for Ash1 and Zelda. Furthermore k_{on} itself is also concentration dependent - in the absence of information on absolute Zelda concentrations, comparison of Zelda binding rates to its dissociation rates is meaningless (Page 8 line 254). In contrast, k_{off} is independent of binding site number and protein concentration; $1/k_{off}$ (residence time) is thus a useful measure. The conclusion that Zelda binding is (on a global level) dynamic, is valid.

We thank the Reviewer for pointing out this issue and fully agree with him on this point. Indeed, the number of Zelda-free binding sites is unknown and therefore k_{on} parameter is impossible to estimate. In this new version, we use FRAP and FCS experiments to only estimate k_{off} and deduce residence times. All k^*_{on} estimates have been removed from the revised manuscript.

I.3.d) The last sentence of the discussion (page 9 line 284): "In this case, we propose that mitotic bookmarking factors prevent the decline to profound states during mitosis." would benefit from rewording to improve clarity ("profound states" is unclear) and focus (is the paper about bookmarking or Zelda?)

We removed this sentence from the discussion section and extended the section on modeling to clarify our conclusions.

Reviewer #2 (II):

II.1. The authors designed different constructs based on the *snaE* enhancer with Zld sites inserted at 5' or 3' end of the enhancer with respect to the promoter and with different numbers of Zld sites. A recent publication (Crocker, Tsai, & Stern. 2017, Cell Rep. 18:287-296.) also explored the role of Zld binding sites in transcriptional activation, observing that increasing Zld sites led to more nuclei showing gene expression but did not change the output levels of gene expression in active nuclei. The authors should discuss the similarities and differences of their findings in relationship to the work published in Cell Reports.

In the original version, we did not write a proper introduction, which explains why we did not discuss Crocker et al important paper. This article is now included in the introduction, page 2 '*Consequently, target gene response is strengthened by Zelda binding, both spatially and temporally, for developmental enhancers⁸ as well as for synthetic enhancers where input parameters are tightly controlled⁹*' and discussed in light of the results of our study in the results section page5 '*In this system, addition of a single CAGGTAG was sufficient..which are not present in the synthetic enhancer examined in⁹*'.

The aim of our study is primarily to decipher the impact of Zelda on temporal dynamics and not on levels of expression. Indeed, while the main strength of the live imaging is to record temporal dynamics, estimating levels of expression from MS2 movies is not straightforward. We feel that a careful analysis concerning the impact of Zelda on levels of expression is beyond the scope of this paper.

However, using a similar approach to what Crocker et al employed (FISH in *nc14*), we examined the impact of extra-Zelda binding sites on the levels of expression (Figure for referee). Consistent with Crocker and colleagues findings, we observe that increasing the number of Zelda binding sites does not affect the intensity of the transcription site. We purposefully did not quantify the intensity haze in the cytoplasm, which would correspond to accumulated mRNAs. Indeed, extra Zelda sites affect the timing of expression and will thus affect the number of accumulated transcripts at the end of *nc14*. Although clear and statistically significant, we do not feel it is appropriate to include this Figure in our manuscript. Indeed, we believe that our study is of quite high standards in terms of quantifications and we are not satisfied with the approximation of levels with a classical FISH approach.

Quantification of absolute levels of expression requires performing single molecule FISH which is still in progress in our team.

Figure: Increasing the number of Zelda sites does not affect nascent spots intensities

A-C) Ventral view of *Drosophila* transgenic embryos at late *nc14* after FISH experiments, carrying a *snaE<MS2* (A), *snaE + 2 Zld<MS2* (B) or *snaE + 3 Zld<MS2* transgene (C). In these maximum intensity projection images transcriptional sites are detected in red with a MS2 probe and nuclei are detected in blue with DAPI. The white arrow indicates the length of the mesodermal pattern. A'-C') Examples of false colored images after nuclei segmentation. Inactive nuclei are colored in grey, active nuclei but not considered in the analysis are in pale blue. The region of

interest (ROI) containing the quantified active nuclei (central region within the presumptive mesoderm) are in bright blue, transcriptional sites are in red. For nuclei segmentation we used an adaptive thresholding algorithm. After segmentation and allocation to nuclei, intensities of each spot within the ROI are quantified. A''-C'') Zoom images of quantified region. (D) Boxplot of transcriptional sites intensities in arbitrary units for the three different genotypes. At steady state (end of *nc14*), we observe that increasing the number of Zelda sites does not alter the strength of transcription, as estimated from nascent spot intensities.

Statistics: *snaE*, N=748 from 8 embryos; *snaE+2Zld*, N=435 from 4 embryos; *snaE+3Zld*, N=348 spots from 3 embryos. (ANOVA, $F_{2, 1528} = 0.927$ and $P > 0.3$).

II.2. Have the authors observed the effect of placing a Zld site in the middle of the enhancer (e.g. as is the case in the Cell Reports paper)? If so, what is its effect on the kinetics of transcription? Is it in between the 5' and the 3' version? Can the authors discuss the impact on Zld site placement (topology) on their ability to speed up transcriptional activation?

Following the useful advice of Reviewer 2, we generated a new transgenic line with an extra Zelda site in the middle of the *sna* enhancer (*snaE+1Zldmid*), schematized in Fig. 1a. Unexpectedly, adding Zelda site in the middle leads to a kinetic of activation, which is not between the 5' and the 3' version. This result is shown in Supplementary Fig. 1b, c and discussed on page 5 of the main manuscript.

II.3. For the modeling work, as shown in Fig. 2E & F and summarized in Table S1, the variations in the a moments (number of transitions) between active and inactive nuclei, specifically between 1Zld 3', 2Zld, and 3Zld, present a less consistent picture compared to the b moments. Some configurations of Zld sites apparently led to near 2-fold changes in the values of a with previously inactive nuclei whereas some configurations of Zld sites yielded larger values of a for active nuclei than inactive nuclei. Adding Zld sites actually led to increases in a more often than not. The apparent interpretation is that some configuration of Zld sites could bypass up to 1 or 2 steps needed for transcriptional activation in inactive nuclei but work against nuclei previously bookmarked through other mechanisms. This means that while Zld may not be fully responsible for the mitotic memory observed at for this particular locus with the *snaE* enhancer, it could still function beyond simply accelerating the kinetics of the individual steps. Alternatively, is this a limitation of the model chosen or are the errors in the fits for a large?

This section has been now entirely reformulated and thoroughly discussed in the text and illustrated with dedicated figures: Fig. 4, Supplementary Fig. 4, Supplementary Table 1 and 2. In particular, the behavior of the ratio of 'a' parameters in inactive and active subpopulations is now taken into account (see response to Ref1, point 1.2.d). This ratio is easier to interpret than the relative changes of 'a' between genotypes, as directly related to the memory bias.

II.4. Have the authors applied the mathematical modeling shown in Fig. 2E and F to their RNAi Zelda experiments to see if Zld depletion leads to a consistent increase in the time spent in each step (the parameter b)?

We thank the referee for this excellent suggestion. We have now estimated the parameters 'a' and 'b' with *RNAi-Zelda* data and these estimates are summarized in Fig. 4 c' and d' and Supplementary Table 1.

We have added a sentence in the main text page 8 : '*Consistent with these findings, upon maternal reduction of Zelda, the 'b' parameter is augmented, when compared to estimates in white RNAi controls (Fig. 4d')*

II.5.a In the FCS analysis, the authors found that Zld binding inside the nucleus is transient and not very stable. However, the authors also noted in the methods that their

fitting model does not take long time correlation into consideration.

We have now introduced a long-lasting time term in our equation to correctly fit the tail of the FCS autocorrelogram. This term was named G^∞ . However, we clearly see that this term is most of the time close to zero (Supplementary Fig. 5f). Therefore, obtaining a correct measure of putative long-time decorrelation (corresponding to a proportion of stably bound Zelda) is impossible by FCS. This is the reason why we also performed FRAP, which is much better suited for slower dynamics.

II.5.b In their live imaging of GFP-Zld (Figure 4H-J), they observed stable loci that contained long-lasting Zld signals (e.g. the blue trace in Figure 4I). The 0.35 s dwell time from the FCS experiments contrasts with their live imaging experiments with loci continuously showing Zld signal over 30+ s. The author should clearly address the possible reasons for this discrepancy. Are the stable loci areas where Zld constantly bind to, showing high levels of activity even though individual binding events are short and unstable? Is it because that their FCS fitting explicitly exclude events longer than 1 s, thereby not including any more stable interactions in their analysis?

It is true that, at first sight, a ≈ 0.35 s dwell time and a ≈ 3 s⁻¹ k_{off} will not be in favour of areas enriched in Zelda. However, as answered in II.5.a, we did not see long time correlations in our FCS measurements performed at random positions inside the nucleus. Therefore, we envisaged performing FCS within Zelda dense regions (hubs), to check if, in these areas, we could observe a third characteristic time, reflecting a much lower k_{off} . Unfortunately, as shown in Supplementary movie 10-12, Zelda hubs are moving quite fast. Thus, an FCS approach would not be able to discriminate which of a binding/unbinding process or of a global collective motion of the hubs in and out of the observation area would contribute to intensity fluctuations.

To overcome this issue, we decided to perform FRAP, since, with the help of imaging during the recovery, we could track the photobleached area and therefore discard the effects due to the collective motion of Zelda dense regions. These new experiments are summarized in Fig. 6 e-g and in Supplementary Fig. 6 c-i and in Supplementary movie 10-12.

Interestingly, we found that the residence time obtained by FRAP in Zelda dense regions was not different from that resulting from FRAP at random positions within the nucleus. We now present these new results in the main text and discuss possible interpretations p13.

See also response to referee3, point **III.15**.

Minor issues in the manuscript:

1. Line 255: The authors refer to a Figure S5D. Should this be Figure S4D?

This is now corrected in the manuscript

2. Line 519-520: The authors describe a 40x water objective with an NA of 1.4. The refractive index of water is only 1.33. Is this a mistake?

We thank the referee for this observation. Indeed, this is a mistake, we corrected it in the text for NA of 1.2 page 22.

3. Figure 4C: The authors should show the result of their fit with the raw FCS to give a sense of the quality of the fit.

A representative fitting of raw FCS is presented in Fig. 5d' and Supplementary Fig. 5e'.

Reviewer #3 (III):

III.1. The influence of Zelda on transcriptional memory is the most convincing and novel aspect of the paper. Why then have a title that only mentions role of Zelda as a temporal coordinator of gene activation (something which has already been highlighted in several other papers)?

We agree that not mentioning memory in the title is unfortunate. However besides memory, we believe that part of the novelty resides in the study of Zelda kinetics and the discovery of Zelda hubs. We therefore replaced the former title by '*Temporal control of gene expression by the pioneer factor Zelda through transient interactions in hubs*'.

III.2. The importance of looking into the dynamics of the binding between Zelda and DNA, and its relevance to the discussion of whether Zelda is a pioneer factor, is not well articulated in the abstract/introduction.

We now included a full introductory section.

III.3. Since transcriptional memory seems very weak in the presence of sufficient Zelda binding sites (and for the intact shadow enhancer), how is memory relevant for embryo development?

This is an important question, discussed in our first report on memory by Ferraro et al 2016 and accompanying Dispatch by J.Chubb. We also wrote a recent review on memory during development that we cite in the revised manuscript (Bellec, Radulescu and Laha Current Opinion in Systems Biology 2018). Indeed, some developmental genes, like endogenous *snail*, may not exhibit a detectable memory because they are activated in a fast manner (thanks to Zelda sites and/or promoter pausing and/or redundant enhancers etc). Nonetheless, this does not mean that all developmental genes will be activated in a fast and synchronous manner (see for example the Doc locus, Supplementary Fig. 2). Note also that an important number of enhancers are not Zelda-dependent and may thus rely on mitotic memory to elicit rapid post-mitotic activation. Unfortunately, until now, evidence of memory on endogenous developmental genes in multicellular embryos is lacking but very likely given new results in ES cells

(Phillips et al., biorxiv doi: <https://doi.org/10.1101/411447>). Our team is actively working on this direction.

III.4. It would be useful to show the data obtained with the intact enhancer in Fig. 1 (schematic representation and number of Zelda binding sites in Fig. 1A, and synchrony curve in Fig. 1M).

As requested, we have now included the schematic representation and synchrony curve of the intact *сна shadow* enhancer in the main Fig. 1I.

Modeling:

III.5. The conclusions drawn from the analysis of the distributions of waiting are weakened by the fact that only one model is considered (based on the assumption that all states have the same average lifetime). This model should be compared to other equally plausible models (e.g. models with a fixed number of states each with a different lifetime).

We thank the Reviewer for this point and agree that alternative models can be envisioned. We however favor the simplest, based on parsimony principle and to avoid overfitting (see methods page 23).

We now discuss a second model where the lifetime of transitions is not kept constant (heterogeneous '*b*') in the Supplementary Methods page 3. As shown in Supplementary Table 2, this model does not increase the quality of the fit and its parameters are uncertain because the model suffers of overfitting. We thus kept most of the discussion with the simpler model of equal '*b*' rates.

See also response to II.3.

III.6. Line 152: "nuclei require at most three transitions to become active". Should it be "rate-limiting" transitions (as the distribution of waiting time would likely only reflect those transitions that are rate-limiting)?

Indeed, the transitions of our stepwise dynamic model could be in principle 'rate limiting'. We now describe the transitions in more detail in the main text, page 8 '*Given the various well characterized steps required prior to productive transcriptional elongation (e.g. promoter opening, Transcription Factor binding, Pre-initiation Complex recruitment), it is reasonable to consider a series of transitions that a nuclei must 'travel' through prior to activation with an allocated duration^{11,24}.*

While we can speculate about the nature of these transitions, their biochemical characterization is beyond the scope of this study.

III.7. Lines 156-159: It seems a bit of a misrepresentation to write that the main difference between nuclei descending from active vs. inactive mothers is seen for parameter *b*, which passes from 3.05 to 2.04, a ~30% decrease, while parameters *a* passes from 208 s to 147 s (also a ~30% decrease).

It is true that the difference between active and inactive involves both variations in 'b' and 'a', but new panels Fig. 4a and 4d show that an increase in 'b' is more consistently seen in nuclei coming from inactive mothers. Furthermore, our new simulations predict, and our data confirm that an increase in 'a' is systematically large in nuclei coming from inactive mothers when 'b' is large (slower transitions). In the revised text we discuss these nuances page 9.

III.8. In Fig 2F, it would be good to also schematize the effect of Zelda on the two populations of nuclei.

In the revised manuscript an entire new Figure is devoted to modeling (Fig. 4) where 5 panels illustrate the effect of Zelda (extra Zelda or Zelda RNAi) on parameters of the model. We feel that schematizing the role of Zelda with our 'stairs' model would be confusing since too many scenarios are possible (for example downward transitions). However, we discuss this in the main text with two paragraphs, page 9.

Dynamic properties of Zelda:

III.9. The authors mention (line 241) that at the end of mitosis Zelda "comes back to the nucleus very rapidly". What does that mean, and how does that compare with how quickly other factors are imported in nuclei after mitosis (e.g. Bicoid, as studied by Gregor et al., 2007)?

On the useful advice of the Reviewer 3, we compared Zelda distribution during the cell cycle to two other DNA binding proteins which are also evicted during mitosis, Pol II phosphor-Serine5 (by immuno-staining) and Bicoid (by live imaging). These new results are summarized in panels b-d of Supplementary Fig. 5 and discussed in the main text page 10 '*When compared to Pol II-Ser5P by immuno-staining....faster than that of Bicoid (Supplementary Fig. 5d).*'

III.10. How come the FRAP and FCS experiments give such different values of the diffusion coefficients?

We thank the reviewer for pointing out this issue. Indeed, our FRAP experimental set-up was not fast enough to measure precisely diffusion coefficients, explaining this discrepancy.

While we first fitted our FRAP data with a 'diffusion only' model, it now clearly appears that it was not a good choice. Therefore, in the revised manuscript, we fitted our FRAP data with a reaction-diffusion model (better suited for a DNA binding protein like Zelda) and only extracted a residence time (not a diffusion coefficient) (Fig. 5c, and Fig. 6f, g). We added explanations regarding this choice in the text page 11, such as '*Because FRAP is not well suited for fast moving proteins, we performed Fluorescence Correlation Spectroscopy (FCS) in living cycle14 embryos and estimated the kinetics properties of Zelda (Fig. 5d, d')*'. We also present the results of fitting with alternative models (one reaction and two reactions), illustrated in Supplementary Fig. 6c-i and discussed page 13 '*Fitting the FRAP experiments with alternative models.....similar to the FCS-extracted residence time (Fig. 5f).*'

III.11. FRAP experiments: The photobleaching duration should be compared to the recovery time (if both photobleaching time and recovery time are of the same order of magnitude, the FRAP experiment does not capture the true dynamics of the fluorophore, due to the “halo effect”). From the methods, it is unclear what the duration of that photobleaching period was, 130 ms or 260 ms (given that two laser pulses are mentioned)? If it was 260 ms, as the recovery time was ~ 0.5 s (Fig. 4B), the halo effect may be artificially decreasing the measured diffusion coefficient.

We now clarified in the methods that the total photobleaching duration is 130ms (page 20). Nevertheless, we definitely agree with the reviewer that this $\sim 1/5$ of the recovery time duration photobleaching implies that this recovery time is a convolution of bleaching and recovery (what the reviewer calls the “halo effect”).

Due to our experimental set-up and our Argon laser power, we could not achieve a shorter photobleaching pulse, that would perturb less the recovery and help in measuring more precisely D_f and k_{off} . Therefore we performed FCS, which had the additional ability to clearly identify the difference between diffusion and exchange characteristic times in our case.

III.12. FCS experiments: The autocorrelation functions shown both for GFP-Zelda (Fig. 4C) and GFP-Ash1 (Fig. S4C) do not fully decay to 0 at the longest times shown. This is a sign of the presence of dynamic processes with characteristic times on the same order of magnitude or larger than the measurement time (10s), and calls into question the validity of the equilibrium model used to fit the data (Eq. 6). The authors mention that the data above 1 s was not analyzed, but this does not eliminate the issue.

The autocorrelation functions do fully decay to zero for both GFP-Zelda and GFP-Ash1 FCS experiments. We apologize if this point was not clear in the original Figure, probably because the $G(\tau)=0$ line was not represented. This is now shown in the new Figure (Fig. 5d' for Zelda and Supplementary Fig. 5e' for Ash1), where instead of showing an average autocorrelation function obtained from all the FCS, we represent a typical FCS curve.

Moreover, we have now introduced a long-lasting time term (G^∞) to our equation (Eq 10, page 22) to correctly fit the tail of the FCS auto-correlogram and show the result in Supplementary Fig. 5f.

See also response to referee II, point II.5a.

III.13.a Since photobleaching is a concern in FCS experiments in live organisms, the authors should report on the excitation intensity, or, at least, on the average number of photons collected per Zelda-GFP.

We thank the reviewer for pointing out this potential issue. Through a direct analysis of each fluorescence fluctuation curve, we removed all FCS curves subject to photobleaching (See also response to referee I, point I.3.b).

Because photobleaching process depends not only on the laser intensity but also on a photobleaching constant that is a property inherent to the molecule and its immediate environment, the best way to visualise the existence of photobleaching is to directly

observe the fluctuations in fluorescence intensity over time (so called fluorescent trace) used to establish the auto-correlation. As long as this fluorescence fluctuates around a constant value, it is considered that no photobleaching occurs during the experiment. Therefore, in the revised Fig. 5d-d', we decided to include an example of a fluorescent trace in a typical FCS experiment, clearly illustrating that, during the observation time, we do not have photobleaching that will affect the correlogram.

III.13.b Can the authors use their FCS data to estimate the concentration of Zelda?

Yes indeed, we can use the FCS to estimate Zelda concentration. Based on the average number of Zelda molecules (N), extracted from the fitted $G(0)$ value following the formula $G(0) = \frac{1}{2^{3/2}N}$ and on the illumination volume, we found that Zelda concentration is around $410^{+/-90}$ nM. We decided not to include this information in the main text but added it in the methods section, page 22.

III.14. The part of the manuscript dealing with the inhomogeneous distribution of Zelda in nuclei seems quite preliminary and almost an after-thought. To be useful, this distribution should be characterized, not just mentioned.

In the revised version, we included a new Figure (Fig. 6), a new Supplementary Figure (Supplementary Fig. 6) and four new Supplementary Movie (Supplementary Movie 9-12) describing Zelda intranuclear distribution at different nuclear cycles. For example, we examined Zelda protein distribution in embryos carrying maternal GFP-Zelda and mCherry-Zelda and found clear bi-color Zelda-hubs. Thus, Zelda hubs are not artefactual aggregates (Fig. 6b). We found that Zelda hubs are more visible at early nuclear cycles and that they are highly dynamic and relatively transient. We also explored the link with transcription (Fig. 6d, Supplementary Fig. 6a, b and Supplementary Movie 11) and studied Zelda dynamics inside hubs (Fig. 6e-g and Supplementary Fig. 6 c-i). Therefore, the part of the revised manuscript on Zelda hubs has been significantly extended. During the revision process, a similar finding has been reported by the lab of M. Eisen (Mir et al., biorxiv 2018).

III.15. An inhomogeneous distribution also raises question as to how the FRAP and FCS experiments were conducted: where they performed in a region of high or low Zelda concentration?

We are grateful to the referee for this interesting suggestion. Given the fast movements of Zelda hubs, performing FCS was not technically possible. Indeed, Zelda hubs movements during FCS acquisition would lead to the generation of long-time correlations. Unfortunately, these could be mis-interpreted as long k_{off} characteristic timings, leading to an overestimated residence time. We therefore performed FRAP on Zelda hubs, since, by tracking the bleached area as long as possible, we cancelled the putative effect of loci displacement to participate to the fluorescence recovery and we therefore mainly analyse the dynamics of Zelda molecules in the loci.

We found that Zelda recovery was as fast as in experiments performed in bulk nuclei. We now show the residence time estimated after FRAP in Zelda hubs and discuss these

results in the manuscript in a new paragraph (pages 11-13) and illustrate it with Fig 6e-g and in Supplementary Fig. 6c-i.

See also response to referee 2, point II.5.b.

Minor comments:

III.16. The sentence “This emphasizes the concept that synchrony is distinct from that of memory” (line 198) is unclear. Maybe because it is grammatically incorrect (what does “that” refer to?). Or maybe because the concept of synchrony has not been well defined. If synchrony means that nuclei all start expressing at the same time (meaning both nuclei from active and inactive mother), then doesn’t it immediately imply that memory is

This sentence has been removed.

III.17. What does “zoomed 6x” (line 492), “4x zoom” (line 446) and “2.1 zoom” (line 429) mean?

It’s the optical zoom applied for imaging settings.

III.18. Fig. 4C: The fit to the FCS data should be shown.

Illustrations of FCS have been changed to include the fitting of a representative raw FCS (Figure 5d’ and Supplementary Fig S5e’).

III.19. line 397: Should be “The circled cluster in panel (H)” (not (J))”.

We highly reorganized this part of the text and we removed this sentence.

III.20. line 520: A water objective should not have a NA above ~1.3.

Thanks for pointing this mistake, we corrected it in the text for NA 1.2.

III.21. line 527: What is “mu” in the definition of tau_Df?

μ is the number of photons in the excitation.

III.22. line 560: The lifetime is the average waiting time.

This paragraph has been re-written.

III.23. Some references are not properly formatted (e.g. ref 24, ref. 27)

This has been corrected.

Reviewer #1 (Remarks to the Author):

The revised manuscript is much improved in all aspects, and the authors have satisfactorily answered most of my concerns. A few minor points remain, which should be revised and would help with clarity, especially for an interdisciplinary readership.

1) Language

The manuscript should be carefully checked for errors of language and grammar. I list here a few examples, but the list is not exhaustive:

Line 70: “we propose that mitotic memory requires long lasting transitions between chromatin states, incompatible with the function of Zelda in accelerating these transitions.”

I would suggest: “...which are accelerated by Zelda, thus overriding mitotic memory of silent states.”

Line 160: “Zelda allows bypassing transcriptional memory”. This is grammatically incorrect. Should be “allows transcriptional memory to be bypassed” Or “Zelda bypasses transcriptional memory”.

Line 310: “Based on the characteristics of pioneer factors, we had expected a role for Zelda in retaining transcriptional memory through mitosis. Thus, our genetic data and modeling indicate that Zelda was not the basis of memory.”

“Thus” should be “However”...

Line 374: “Hence Zelda hubs have been observed very recently with other methods”. “Hence” does not make sense here. I suggest, “Consistent with this,” Or “

Line 735: “where each tracked nuclei is given a random colour” Should be “nucleus”.

2) Zelda and transcriptional activation

My comments on Zelda and transcriptional activation have mostly been addressed, with the following exceptions:

2.1) Figure 1g' and legend (Line 742), is stated as showing: "Representative image exhibiting the spatial

domain (grey, here 25 μ m surrounding the ventral furrow) defined by precise D/V coordinates."

However, in Supplementary Figure 1A this domain appears to be about 100 μ m.

In main text: Line 112, it is stated: "Unless otherwise indicated, we studied temporal dynamics of gene activation in a region of 50 μ m centered around the ventral furrow."

It would make more sense to show the 100 μ m box, with 50 μ m each side of the furrow, in all figures.

For clarity, indicate grey zone in Fig 1g', on Fig 1g, or add scale bar.

2.2) Figure S1d. The x – axis is labelled "Distance from gastrulation" – this is unclear, as gastrulation is a developmental event rather than a specific place in the embryo. "Ventral furrow" as used elsewhere would be better. Fig S1d legend "gastrulation line" – change to "ventral furrow".

2.3) Definition of the domain in which imaging was performed. There are several instances in the manuscript in which the domain that was imaged is referred to as a "spatially defined pattern". This may be a language issue: when I read "pattern" I expect a pattern within a domain. "Spatially defined domain: would be more accurate.

E.g., Line 119:

"...but also the temporal coordination among a spatially defined pattern (i.e the presumptive mesoderm, 50 μ m around the furrow)," Change to e.g, .."among nuclei in a spatially defined domain".

Line 124

“The precise kinetics of gene activation, i.e synchrony curves, was quantified as a percentage of active nuclei within a defined spatial pattern for each transgene during the first 30min of nc14.”
Change to e.g., “within a defined spatial domain”

2.4) Line 154. “...boosted the spatio-temporal response to the dorso-ventral gradient (Supplementary Fig. 1d).”

I still have difficulty with this statement. Upon reading this sentence, I expect to see in Figure 1d, that the kinetics or amount of activation change with distance from the ventral furrow, and that this is in some way related to an existing gradient, but I do not see that in the data. For any given transgene it looks to me as if the response does not change with distance from the furrow. If the authors wish to highlight a spatial component, then the difference (if any) should be pointed out and statistics provided. The “dorso ventral gradient” to which the transgenes are responding, should also be presented graphically. Otherwise I suggest: “led to more rapid activation across the entire spatial domain.”

2.5) In the rebuttal letter, the authors state that Twist and Dorsal are non-limiting in the zone analysed. This should be stated in the main text, and the Dorsal gradient should be explained or shown in a diagram for those not familiar with it.

3) Bookmarking and modelling.

The description of the model and conclusions is much clearer, only few minor points remain.

3.1) Line 252: “The model predicts that the distribution of waiting times prior to activation for a given gene in a set of nuclei can be described as a mixture of gamma distributions, in which the mixing occurs over the shape parameter...”

a) This seems to be a model feature rather than a prediction. E.g., “in the model, the distribution... is described as..”

b) Briefly define gamma distribution and the contribution of the shape and scale parameters. Very few biologists will be familiar with this term. (Just as most mathematicians and many biologists will not know the shape of the Dorsal gradient).

3.2) Line 257: “The model also predicts that the average number of steps (parameter ‘a’) can be roughly computed from the first two moments of the distribution of waiting times (Eq.S3 Supplementary Methods).”

Again, this is a feature and not a prediction of the model. I would suggest, again: “using the model, the ... can be computed...”

3.3) Line 288: “In order to gain some insights, we have used numerical simulations of an extended version of our model that includes modified transition dynamics during mitosis.”

This extended model is less well explained in the main text than the first model. Its conclusions are equally important so it should be explained better in the main text.

The conclusions of the modeling are now very clear.

4) Zelda dynamics

4.1) The part on Zelda dynamics is now arranged in a more logical order and the additional characterisation of “Zelda hubs” is nice. The authors report the finding that there seems to be no difference in Zld kinetics in the hubs or out of them, and no detectable difference in transcription of PolII localisation.

It therefore comes as a surprise that the authors conclude later:

Line 434: “Our data support a model whereby Zelda binds transiently to chromatin in localized nuclear microenvironments to accelerate the various transitions required prior to transcriptional activation (e.g recruitment of transcription factors, recruitment of Pol II and general transcription factors).

There is no evidence that the nuclear microenvironment has any impact on transcription. In fact rather the contrary, which is interesting and should be discussed.

4.2) The new title places emphasis on hubs, which I think does not do justice to the very nice work in the rest of the paper. The authors may wish to consider revising the title.

Reviewer #2 (Remarks to the Author):

The authors conducted additional experiments (using a new line of flies (Zld in the middle), FRAP, etc.) and expanded their modelling analysis to address questions raised by the reviewers. In addition, their revised introduction now provides better coverage of the prior art and their rationale for performing live imaging experiments. In my opinion, they have satisfactorily answered all of the concerns raised. I therefore recommend that this manuscript be published.

We are grateful to Reviewer 1 for his/her comments.
Based on his/her advice, we modified our manuscript. Below we provide a detailed, point-by-point account of the changes in the revised manuscript.

Reviewer #1 (Remarks to the Author):

1) Language

The manuscript should be carefully checked for errors of language and grammar.
We carefully checked our manuscript for language and grammar.

Line 70: “we propose that mitotic memory requires long lasting transitions between chromatin states, incompatible with the function of Zelda in accelerating these transitions.”

I would suggest: “...which are accelerated by Zelda, thus overriding mitotic memory of silent states.”

Replaced

Line 160: “Zelda allows bypassing transcriptional memory”. This is grammatically incorrect. Should be “allows transcriptional memory to be bypassed” Or “Zelda bypasses transcriptional memory”.

Replaced by ‘Zelda bypasses transcriptional memory’

Line 310: “Based on the characteristics of pioneer factors, we had expected a role for Zelda in retaining transcriptional memory through mitosis. Thus, our genetic data and modeling indicate that Zelda was not the basis of memory.”

Replaced

“Thus” should be “However”...

Replaced

Line 374: “Hence Zelda hubs have been observed very recently with other methods”. “Hence” does not make sense here. I suggest, “Consistent with this, ...” Or “

Replaced by ‘Consistent with this...’

Line 735: “where each tracked nuclei is given a random colour” Should be “nucleus”.

Replaced

2) Zelda and transcriptional activation

2.1) Figure 1g' and legend (Line 742), is stated as showing: "Representative image exhibiting the spatial domain (grey, here 25 μ m surrounding the ventral furrow) defined by precise D/V coordinates."

However, in Supplementary Figure 1A this domain appears to be about 100 μ m.

In main text: Line 112, it is stated: "Unless otherwise indicated, we studied temporal dynamics of gene activation in a region of 50 μ m centered around the ventral furrow."

It would make more sense to show the 100 μ m box, with 50 μ m each side of the furrow, in all figures.

For clarity, indicate grey zone in Fig 1g', on Fig 1g, or add scale bar.

We have now modified Fig. 1m and Supplementary Fig. 1a to indicate the analyzed region in μ m.

2.2) Figure S1d. The x – axis is labelled "Distance from gastrulation" – this is unclear, as gastrulation is a developmental event rather than a specific place in the embryo. "Ventral furrow" as used elsewhere would be better. Fig S1d legend "gastrulation line" – change to "ventral furrow".

Replaced

2.3) Definition of the domain in which imaging was performed. There are several instances in the manuscript in which the domain that was imaged is referred to as a "spatially defined pattern". This may be a language issue: when I read "pattern" I expect a pattern within a domain. "Spatially defined domain: would be more accurate.

We agree and replaced all 'spatially defined pattern' by 'spatially defined domain'.

E.g., Line 119:

"...but also the temporal coordination among a spatially defined pattern (i.e the presumptive mesoderm, 50 μ m around the furrow)," Change to e.g. .."among nuclei in a spatially defined domain".

Changed

Line 124

"The precise kinetics of gene activation, i.e synchrony curves, was quantified as a percentage of active nuclei within a defined spatial pattern for each transgene during the first 30min of nc14." Change to e.g., "within a defined spatial domain"

Done

2.4) Line 154. "...boosted the spatio-temporal response to the dorso-ventral gradient (Supplementary Fig. 1d)."

I still have difficulty with this statement. Upon reading this sentence, I expect to see in Figure 1d, that the kinetics or amount of activation change with distance from the ventral furrow, and that this is in some way related to an existing gradient, but I do not see that in the data. For any given transgene it looks to me as if the response does not change with distance from the furrow. If the authors wish to highlight a spatial component, then the difference (if any) should be pointed out and statistics provided. The "dorso ventral gradient" to which the transgenes are responding, should also be presented graphically. Otherwise I suggest: "led to more rapid activation across the entire spatial domain."

We changed the text as suggested.

2.5) In the rebuttal letter, the authors state that Twist and Dorsal are non-limiting in the zone analysed. This should be stated in the main text, and the Dorsal gradient should be explained or shown in a diagram for those not familiar with it.

We have now added a schematic of dorsal gradient, Supplementary Figure 1b

3) Bookmarking and modelling.

The description of the model and conclusions is much clearer, only few minor points remain.

3.1) Line 252: "The model predicts that the distribution of waiting times prior to activation for a given gene in a set of nuclei can be described as a mixture of gamma distributions, in which the mixing occurs over the shape parameter..."

a) This seems to be a model feature rather than a prediction. E.g., "in the model, the distribution... is described as..."

We agree with the comment and remove 'predicts' in the main text: *In the model, the distribution of waiting times prior to activation for a given gene in a set of nuclei can be described as a mixture of gamma distributions.*

b) Briefly define gamma distribution and the contribution of the shape and scale parameters. Very few biologists will be familiar with this term. (Just as most mathematicians and many biologists will not know the shape of the Dorsal gradient).

We modified the text clarify this point: *Gamma distributions are frequently used in statistics for modeling waiting times. These distributions depend on two parameters, the shape parameter 'a' and the scale parameter 'b'. When, like in our model, the waiting time is the sum of a number of independent, exponentially distributed steps of equal mean duration, 'a' is the number of transitions (steps) while 'b' is the mean duration. Thus, 'a'=1 corresponds to the exponential distribution. A mixture of gamma distributions covers the case when the number of transitions (parameter 'a') is random.*

3.2) Line 257: “The model also predicts that the average number of steps (parameter ‘a’) can be roughly computed from the first two moments of the distribution of waiting times (Eq.S3 Supplementary Methods).”

Again, this is a feature and not a prediction of the model. I would suggest, again: “using the model, the ... can be computed...”

Changed

3.3) Line 288: “In order to gain some insights, we have used numerical simulations of an extended version of our model that includes modified transition dynamics during mitosis.”

This extended model is less well explained in the main text than the first model. Its conclusions are equally important so it should be explained better in the main text.

we have now explained the extended model: In order to gain some insights, we have used numerical simulations of an extended version of our model that includes modified transition dynamics during mitosis. In this version, we consider that at the beginning of mitosis, the states of active and inactive mother nuclei are OFF1 and OFF3, respectively. During mitosis, nuclei can undergo reversible (upward and backward) transitions. After mitosis, the resulting daughter nuclei follow the irreversible transition scheme represented in Fig. 4a. The simulations predict that the bias in ‘a’ values (difference in the number of steps to reach active state, evaluated by $a_{inactive}/a_{active}$ and referred to as memory bias) is correlated to the ‘b’ values (Supplementary Methods).

The conclusions of the modeling are now very clear.

4) Zelda dynamics

4.1) The part on Zelda dynamics is now arranged in a more logical order and the additional characterisation of “Zelda hubs” is nice. The authors report the finding that there seems to be no difference in Zld kinetics in the hubs or out of them, and no detectable difference in transcription of PolII localisation.

It therefore comes as a surprise that the authors conclude later:

Line 434: “Our data support a model whereby Zelda binds transiently to chromatin in localized nuclear microenvironments to accelerate the various transitions required prior to transcriptional activation (e.g recruitment of transcription factors, recruitment of Pol II and general transcription factors).

There is no evidence that the nuclear microenvironment has any impact on transcription. In fact rather the contrary, which is interesting and should be discussed.

We agree that the link between Zelda hubs and transcription is not clear and this is why we clearly stated in the results of our main text p12: '*connecting transcriptional activation to Zelda hubs would require broader analysis with adapted imaging methods.*'

We are currently trying to decipher this question.

As suggested by the referee, we however rephrased the last paragraph of our manuscript, distinguishing what our data support, from what we propose as an open pending question.

*'Our data support a model whereby Zelda binds transiently to chromatin in localized nuclear microenvironments; to **potentially** accelerate the timing of the transitions required **prior** to transcriptional activation (e.g. **local chromatin organization**, recruitment of transcription factors, recruitment of Pol II and general transcription factors).'*

4.2) The new title places emphasis on hubs, which I think does not do justice to the very nice work in the rest of the paper. The authors may wish to consider revising the title.

We thank the referee for the compliment of the full paper and we understand her/his point. Indeed, we hesitated a lot on whether to change the title or not.

Our preprint is on Biorxiv and we already changed once based on referee 3 advice. Therefore, we decided to stick with the actual one.